



# Air pollution trapping in the Dresden Basin induced by natural and urban topography

Michael Weger[1] and Bernd Heinold[1]

[1]Leibniz Institute for Tropospheric Research, Leipzig, Germany

**Correspondence:** Michael Weger (weger@tropos.de)

**Abstract.** The microscale variability of urban air pollution is essentially driven by the interaction between meteorology and urban topography, which remains challenging to represent spatially accurately and computationally efficiently in urban dispersion models. Natural topography can additionally exert a considerable amplifying effect on urban background pollution, depending on atmospheric stability. This requires an equally important representation in models, as even subtle terrain-height variations can enforce characteristic local flow regimes. In this model study, the effects of urban and natural topography on the local winds and air pollution dispersion in the Dresden Basin in the Eastern German Elbe valley are investigated. A new, efficient urban microscale model is used within a multiscale air-quality modeling framework. The simulations that consider real meteorological and emission conditions focus on two periods in late winter and early summer, respectively, as well as on black carbon (BC), a key air pollutant mainly emitted from motorized traffic. As a complement to the commonly used mass concentrations, the particle-age content (age concentration) is simulated. This concept, which was originally developed to study hydrological reservoir flows in an Eulerian framework, is adapted here for the first time for atmospheric boundary-layer modeling. The approach is used to identify stagnant or recirculating orographic air flows and resulting air pollution trapping. An empirical orthogonal function (EOF) analysis is applied to the simulation results to attribute the air pollution modes to specific weather patterns and quantify their significance. Air quality monitoring data for the region are used for model evaluation. The model results show a strong sensitivity to atmospheric conditions, but generally confirm increased BC levels in Dresden due to the valley location. The horizontal variability of mass concentrations is dominated by the patterns of traffic emissions, which overlay potential orography-driven pollutant accumulations. Therefore, an assessment of the orographic impact on air pollution is usually inconclusive. However, using the age concentration metric, which filters out direct emission effects, previously undetected spatial patterns are discovered that are largely modulated by the surface orography. The comparison with a dispersion simulation assuming spatially homogeneous emissions also proves the robustness of the orographic flow information contained in the age concentrations distribution and shows it to be a suitable metric to assess orographic air-pollution trapping. The simulation analysis indicates several air quality hotspots on the south-western slopes of the Dresden Basin and in the southern side valley, the Döhlen Basin, depending on the prevailing wind direction.



# 1 Introduction

The embedding surface orography exerts besides the urban microscale structure, like the distribution of buildings and land use, an important influence on the dispersion of urban air pollutants (Wallace et al., 2010; Zeeman et al., 2022). Through a combination of thermal and mechanical effects, complex terrain induces local flow regimes, which can dominate the transport of pollutants under a weak synoptic forcing. For example, horizontal potential-temperature gradients resulting from the inclination of the shallow surface layer drive up-slope winds during surface warming and down-slope (katabatic) winds during

surface cooling (Whiteman, 1990). The vertical circulations associated with slope winds significantly alter the local boundary layer structure. Up-slope winds tend to inhibit the height of a daytime convective boundary layer, as subsidence is enforced from mass continuity over the valley center (Weigel et al., 2006). Down-slope winds, on the other hand, play an important role in the formation of diurnal cold-air pools (CAP). Within sheltered valleys or behind valley constrictions, the boundary layer thickens from the accumulation of relatively dense air advected by the down-slope flows over time (Whiteman et al., 1996; Fast

et al., 1996). Due to the increased stratification, down-slope winds may eventually detach from the surface before reaching the valley floor. The resulting flow separation drastically reduces vertical mixing across the CAP boundaries. The air within the CAP then becomes stagnant or can only recirculate horizontally. In fact, Chemel and Burns (2015) found a significant dependence of dispersion characteristics on emission height in their numerical simulation of a developing CAP within an idealized valley cross-section. Tracers emitted above the side slopes were effectively diluted by the slope-flow circulations. Pollutants

emitted near the valley floor remained largely trapped within a layer of roughly $100\,\mathrm{m}$ depth and accumulated over time. The potential for a given valley geometry to pool cold air is closely connected to the variation of the valley cross-section along the valley axis (McKee and O'Neal, 1989). A widening cross-section towards the valley exit results in a differential cooling rate that forces near-surface draining currents fed by the converging down-slope flows and subsidence over the valley center. Such draining currents can play an important role in the transport of air pollution from higher-elevated tributary valleys towards a

lower-elevated basin, as, e.g., studied by Sabatier et al. (2020) for a French alpine valley system. The presence of urban areas further modifies orographic wind systems in a presumably complex way. Rendón et al. (2020) carried out numerical experiments to investigate the interaction of the urban heat island (UHI) effect with the slope-flow circulation. They distinguished urban from rural areas by both a larger surface heat flux and surface-roughness length for the urban areas. Depending on valley width, the slope-flow circulations competitively interacted against the UHI circulation during the daytime and interacted

amplifying during nighttime. The spatial distribution of modeled urban air pollution responded to the sign of this interaction, as pollutants concentrated more over the side slopes during competitive interaction. While the idealized valley geometry used by Rendón et al. (2020) is quite distinct, interactions of the UHI with topography and topography-related wind systems can also be observed for an orography as subtle as the Seine valley near Paris, where height differences are in the range of only $50\,\mathrm{m}$ (Troude et al., 2001). This suggests that surface orography can have a significant effect on the air quality in many urban

areas. Nevertheless, subtle surface height variations are often neglected in urban dispersion simulations at the microscale. The majority of simulation studies concerned with the dispersion of air pollutants in complex terrain used idealizations with respect to orography (Lehner and Gohm, 2010; Chemel and Burns, 2015), or meteorological boundary conditions and emissions (Ken-





jereš and Hanjalić, 2002; Sabatier et al., 2020), as their aim was to explore more specific aspects of the pollutant transport. In any case, a more general illumination of orography-related effects in the framework of a realistic dispersion simulation case study still remains to be carried out. There is also a potential for practical implications of such a study: As elaborated, the impact of emissions can be much more far-reaching than the areas located in direct proximity of the sources. More specifically, certain orographic regions are predestined to act as secondary pollution sources under the right meteorological conditions. The detection of such regions, which were likely previously not recognized from their inconspicuousness or/and from a lack of air-quality monitoring, may aid to direct more research to such areas with the ultimate goal to inform air-quality policy in the future.

The temporal variability of air-pollution concentrations from local sources is governed by two main factors: The variability in the emission strength and the variability in mixing/ventilation efficiency of the atmospheric flow constrained by the topography. An additional characterization of local air pollution can be based on the aerosol age, as this quantity allows to differentiate between both contributions, in contrast to an analysis based on the particle mass concentration alone. Indeed, in areas with stagnant or recirculating air flows, both a high particle mass and age are to be expected, while in emission-dominated areas particle age is not expected to be increased. The obvious tools to simulate both quantities simultaneously are probably the Lagrangian particle dispersion models (LPDM), as by design they allow tracking of arbitrary properties attached to the transported particles in space (Roberto et al., 2017; Kleinman et al., 2003; Pisso et al., 2019). LPDMs, however, have their own strengths and weaknesses. For example, the spatial resolution in LPDMs is not uniform, allowing for a computationally efficient dispersion simulation of point sources. At the same time, a fixed particle count does not always guarantee a sufficient spatial resolution in every part of the computation domain. Moreover, the derivation of continuous fields from particle counts requires further post-processing using a suitable algorithm (box counting, kernel functions). An alternative approach for age computations is provided with the Eulerian-based tracer methods, which originally have been developed to estimate stratospheric and oceanic transport timescales in general circulation models. Hall and Plumb (1994) first identified Green's function of the stationary continuity equation with the age spectrum, from which the mean age follows by computing the first moment. In practice, the mean age of stratospheric age can be inferred from a passive tracer simulation using an impulsive boundary condition for the troposphere (Hall and Waugh, 1997). Instead of a delta pulse, also a linearly-increasing boundary condition can be used. The mean age is then given by the mixing-ratio time lag between a stratospheric location and the boundary condition. Boering et al. (1996) showed this method to be equivalent to subjecting an age tracer to an emission rate of unity and zero boundary conditions, which was also the method of England (1995) to estimate the age of water in the world oceans. Finally, in Delhez et al. (1999) and Deleersnijder et al. (2001) an Eulerian theory for particle age is presented, which includes an evolving constituent mass concentration with source and sink terms as a further generalization to the aforementioned age tracer method for fluids. Deleersnijder et al. (2001) define the mean (mass-weighted) particle age as the quotient of the so-called age concentration and the constituent mass concentration. Since the age concentration adheres to the same transport equation (with the exception of the source term) as the mass concentration, it provides a straightforward and numerically efficient way to implement age computations in an existing Eulerian dispersion model. While the methodology of Deleersnijder et al. (2001) was primarily targeted at hydrological flows, it can be naturally adapted in atmospheric dispersion modeling as already demonstrated by Han





and Zender (2010) for a global-scale desert dust simulation. To the author's knowledge, the age-concentration methodology has
not yet been applied in atmospheric boundary-layer modeling, where we propose its usefulness given the analogy of orographic
air flows to the already studied hydrological reservoir flows (Mercier and Delhez, 2010; Zhang et al., 2010).

In this paper, LES microscale simulations with the topography-resolving urban dispersion model CAIRDIO (Weger et al.,
2021) are performed and analyzed to (1) propose the age-concentration as a metric for the assessment of the topography ef-
fect on air-pollution dispersion, and (2) to characterize the flow dynamics in the Dresden basin in a novel way to find a link
between meteorology and local air-pollution exposure. The Dresden Basin is a widened section of the Elbe Valley located in
the east of Germany. It is approximately $45\,\mathrm{km}$ long and $10\,\mathrm{km}$ wide and contains the mid-sized city of Dresden with around
half a million inhabitants. The combination of urban emissions and the enclosure of the city area by plateaus and hills approxi-
mately $300\,\mathrm{m}$ high to the south and north makes the area susceptible to orographic air-pollution trapping under a weak synoptic
forcing. However, terrain height variations can be regarded as only moderate compared to, for example, some much steeper
Alpine valleys studied in a similar framework (Harnisch et al., 2009). This study expands upon an earlier simulation case study
presented in Weger et al. (2022a) by focusing on the natural topographic impact on the dispersion of local black carbon (BC)
emissions. In Weger et al. (2022a) the effect of urban topography on the intra-urban variability of BC and particulate matter
was investigated. Therein, natural topography was technically represented but played no significant role as it was basically flat.
In this study, urban topography is again explicitly represented as before in order to also include synergetic effects between both
types of topography for a most realistic simulation of the meteorology in the Dresden Basin. This study relies on an extended
total simulation period of 24 days for more representative results, which is split into a late winter and an early summer period
of equal length. Furthermore, the Eulerian-based simulation of particle age provides a significant novelty in the framework of
a regional air-quality study. The paper is structured as follows: Section 2 introduces the Eulerian framework for particle-age
computations alongside an application of the transport equations on a simplified basin model in order to illustrate the approach.
It furthermore contains a brief technical description of the numerical dispersion model CAIRDIO, the simulation setups and
model runs. In Section 3, the dispersion simulation results are presented together with a meteorological description of the two
simulation periods and a principal component analysis to identify the most important weather and dispersion patterns, which
are illuminated from multiple perspectives. Finally, Section 4 provides a conclusion and summary.

## 2  Methodology

### 2.1  Age-concentration dispersion modeling

#### 2.1.1  Set of governing equations

According to the age-averaging hypothesis outlined in Delhez et al. (1999) and presented in more detail in Deleersnijder et al.
(2001), the age $a$ of an ensemble of particles is defined as the mass-weighted average of the individual particle ages within
that ensemble. The term particle can be understood here in a more general way as an element of a hypothetical subgrid-scale
decomposition of the regarded constituent mass $m$ (e.g. of an air pollutant) contained within a given control volume. While the




age itself is an intensive quantity, the age content $A = a \cdot m$ shares the same additive property as particle mass does, i.e. the age content of two combined ensembles is the sum of its age contents. Based on this additive principle, Deleersnijder et al. (2001) derived a transport equation for the age-content mixing ratio $q_a$. The age concentration $\alpha$ follows by multiplying $q_a$ with the reference density of air $\rho_0$. The two transport equations for $\alpha$ and $c$ are given by:

$$\partial_t c = -\nabla \cdot (\boldsymbol{u} c) + \nabla \cdot \left( k_e \rho_0 \nabla \frac{c}{\rho_0} \right) + e - \tau c \tag{1}$$

$$\partial_t \alpha = -\nabla \cdot (\boldsymbol{u} \alpha) + \nabla \cdot \left( k_e \rho_0 \nabla \frac{\alpha}{\rho_0} \right) + c - \tau \alpha \tag{2}$$

The velocity vector is denoted by $\boldsymbol{u}$, the eddy diffusivity by $k_e$, the inverse deposition time scale by $\tau$, and the emission rate by $e$ [$\mu g\,m^{-3}\,s^{-1}$]. Note that $\nabla \cdot (\rho_0 \boldsymbol{u}) = 0$ using the anelastic approximation for an incompressible fluid. The striking similarity of Eq. 2 with Eq. 1 implies that the numerical schemes already used for the computation of $c$ can be also used for the age computations. Different are only the terms without spatial derivatives, i.e. the source and sink terms. Eq. 2 represents

not the most general form presented in Deleersnijder et al. (2001) but implicitly assumes that particles have an age of zero at the instant of emission. Thus in our framework, particle age refers to the elapsed time from the instant of particle entrance into the computation domain, whether it be from emissions or transport across the lateral boundaries. Having computed the solution of the coupled transport equations, the spatial distribution of the average age is obtained according to the age-averaging hypothesis:

$$a = \frac{c_a}{c}. \tag{3}$$

Eqs. 1 - 2 can be also cast into a transport equation for $a$:

$$\partial_t a = -\frac{1}{\rho_0} \nabla \cdot (\rho_0 \boldsymbol{u} a) + \frac{1}{\rho_0} \nabla \cdot (k_e \rho_0 \nabla a) + \frac{2 \rho_0 k_e}{c} \nabla \frac{c}{\rho_0} \cdot \nabla a - e \frac{a}{c} + 1 \tag{4}$$

Due to additional numerical difficulties, it is, however, not recommended to solve Eq. 4. Nevertheless, Eq. 4 provides some insight into the different processes influencing age. The first two terms on the right-hand side of Eq. 4 represent the advective-
diffusive transport of $a$. The third term arises from the mass-weighting effect in the diffusive fluxes for $a$, as the spatial values of $a$ associated with larger values of $q = c/\rho_0$ bear a higher significance. Hence, the term has the largest contribution when the diffusive fluxes for $a$ and $q$ are orientated in the same direction. It vanishes when the fluxes are orientated perpendicular to each other, or alternatively $\nabla q = 0$. The fourth term describes the rejuvenating effect of freshly-emitted particles with zero age. Note that, on the other hand, deposition of $c$ has no effect on $a$, as all particles are assumed to share the same probability
to deposit irrespective of their age. Finally, the last term represents the aging process itself, which proceeds at the rate of time advancement.



### 2.1.2 A simplified model of a basin

In order to illustrate the significance of the age concentration in the study of air-pollution trapping within orographic depressions, Eqs. 1-2 are applied on a semi-enclosed reservoir considering a highly simplified spatial concentration distribution and
exchange processes only. In Fig. 1a, a 2-D model of a basin is sketched, in which the basin states for $c_b$ and $\alpha_b$ are assumed to be approximately spatially homogeneous from the effect of internal mixing processes. An air-pollutant emission flux from the basin floor is represented by $e$, while particle deposition occurs at a rate proportional to an inverse deposition time scale $\tau$. Above the basin side walls, synoptic-scale winds (denoted by $v$) prevail. Turbulent motions from the resulting wind shear between the sheltered basin atmosphere and the winds above drive vertical concentration fluxes at a rate proportional to $\kappa\,[\mathrm{s}^{-1}]$.
The concentration and age concentration within the free troposphere are denoted by $c_t$ and $\alpha_t$, respectively. To spatially discretize Eqs. 1-2 for this simplified configuration, it is sufficient to consider 4 jointed grid boxes as depicted in Fig. 1b. Box "b" represents the basin atmosphere, box "t" the volume directly above and boxes "l" and "r" the states adjacent to the left and right of volume "t", respectively. The prevailing wind direction is from the left, which results in a positive advective velocity $v_a\,[\mathrm{s}^{-1}]$. The state in box "l" is given by $c_l$ and $\alpha_l$. For each of the remaining 3 boxes, two ordinary differential equations for
the unknowns $c$ and $\alpha$ can be written, respectively:

$$\dot{c}_t = v_a(c_l - c_t) + \kappa(c_b - c_t) \tag{5}$$

$$\dot{\alpha}_t = v_a(\alpha_l - \alpha_t) + \kappa(\alpha_b - \alpha_t) + c_t \tag{6}$$

$$\dot{c}_b = e + \kappa(c_t - c_b) - \tau c_b \tag{7}$$

$$\dot{\alpha}_b = \kappa(\alpha_t - \alpha_b) - \tau \alpha_b + c_b \tag{8}$$

$$\dot{c}_r = v_a(c_t - c_r) \tag{9}$$

$$\dot{\alpha}_r = v_a(\alpha_t - \alpha_r) + c_r \tag{10}$$

Here, an upwind formulation was used for horizontal advection. To simplify matters further, the stationary states are considered only, and it is furthermore neglected deposition. Then, the solutions of the set of equations are given by:





$$c_t = c_l + \frac{e}{v_a} \tag{11}$$

$$\alpha_t = \alpha_l + \frac{2c_l}{v_a} + e \left( \frac{\kappa + v_a}{v_a^2 \kappa} + \frac{1}{v_a^2} \right) \tag{12}$$

$$c_b = c_l + e \left( \frac{1}{\kappa} + \frac{1}{v_a} \right) \tag{13}$$

$$\alpha_b = \alpha_l + \frac{2c_l}{v_a} + \frac{c_l}{\kappa} + e \left( \frac{(\kappa + v_a)^2}{v_a^2 \kappa^2} + \frac{1}{v_a^2} \right) \tag{14}$$

$$c_r = c_t \tag{15}$$

$$\alpha_r = \alpha_t + \frac{c_r}{v_a} \tag{16}$$

At this point, one can distinguish two principal cases. In case one, the concentration inflow into the domain is solely represented by the emission source $e$, thus $c_l = 0$, and $\alpha_l = 0$. It is furthermore assumed $v_a >> \kappa$, i.e. horizontal advective transport above the basin occurs at a much faster rate than the vertical transport across the basin top. In this case, the concentration within the basin is approximated by $c_b = e/\kappa$, and the age concentration by $\alpha_b = e/\kappa^2$. This describes the trivial case when the air pollutant concentration increases within the boundary layer as a result of a more stable stratification. Note that in this case, the mean residence time is $a = 1/\kappa$, which in practice can be many hours. Air pollutants can therefore accumulate over a prolonged period of time, provided that the stable stratification persists. The age concentration does not behave fundamentally differently in this regard, as it also tends towards infinity for $\kappa \to 0$. However, it increases proportionally to the power of two of $a$. Consequently, horizontal variations in $\kappa$, which may be modulated to a large extent by the surface orography, can be better discriminated in a map of $\alpha$ compared to a map of $c$. Case two represents a concentration inflow from a source located outside the basin, which is represented by $e = 0$, $c_l > 0$, and $\alpha_l > 0$. It is clear from the diffusion law that in the absence of a source the concentration within the basin $c_b$ can only approach the external value $c_t$, respective $c_l$. In this regard, the age concentration behaves fundamentally differently. In fact, the term $c_l/\kappa$ in Eq. 14 tends again to grow beyond all boundaries for an increasingly stable stratification, similar to case one. This reflects the relative insignificance of the exact location of the emission source in the spatial distribution of $\alpha$, a property which also follows immediately from the structure of Eq. 2 (i.e. no explicit emission term appears therein). It is also interesting to note the exerted effect of the basin on the age concentration in the atmospheric layer above. For sole horizontal advection, $\alpha$ does not remain constant in time, but increases due to particle aging (represented by Eq. 16). Note that for grid cell "t", which communicates with the basin below, the aging rate is exactly the double of the aging rate in cell "r". This reflects the possibility of particles temporarily leaving the horizontal stream to halt in cell "b". This effectively reduces the transport speed by a factor of two, which is irrespective of the value of $\kappa$. Only a hard physical barrier between cells $b$ and $t$ ($\kappa = 0$) eliminates this effect. Thus, the downwind influence of the basin on $\alpha$ in case two can be understood by a history of an effectively accelerated aging rate for the time the advected air parcel needed to traverse the basin cross-section. For significant wind speeds or slim basin cross sections, this effect can be, however, considered to be irrelevant.





To finally graphically illustrate the remarks given above, Eqs. 1-2 are integrated numerically for an idealized basin cross-section to compare the two contrary cases. The detailed simulation setup with the model CAIRDIO is described in Appendix B. Fig. 2a,c, and e show the distribution of $c$, $\alpha$, and the mean particle age $a$ for an emission source placed at the bottom of the basin. In this case, the recirculation effect of the emitted tracer is reflected both in $c$ and $\alpha$, as both fields show a significant accumulation within the pot-shaped basin. In Fig. 2b the emission source itself is visible by a quite sharp maximum, which is not seen in Fig. 2d. The associated mean particle age $a$ (Fig. 2f) shows a complementary distribution to $c$ in the sense that the emission source and the surrounding volume are characterized by a minimum and lower than average values, respectively. This is a result of the diluting effects of freshly emitted particles with zero age. Next, Fig. 2b,d, and f show the second case, where the emission source is located on the plateau above the left sidewall. As the emissions are also allowed to vertically disperse into faster-moving layers of air, the concentration generally tends to decrease with increased downwind distance from the source. Some of the emissions entering the basin are trapped within the recirculation zone covering the lower half of the basin. As a result, the distribution of $c$ within the basin is quite homogeneous (which reflects $c_b \approx c_t$). In contrast to $c$, the distribution of $\alpha$ (Fig. 2c) is fundamentally different, as the emission source is not directly visible, but instead the effect of the recirculating air within the basin is reflected by the accumulation of $\alpha$. Note also that in this case the distribution of $\alpha$ and $a$ have a quite similar appearance (as opposed to $c$ and $\alpha$ in case one). More generally, of the set of the three depicted variables, only $\alpha$ shows a consistent spatial pattern among the two contrary cases, as indeed the patterns in Fig. 2c and d appear quite similar. While the spatial distributions of $c$, as well as $a$ are dominated by the emissions, the spatial distributions of $\alpha$ show an accentuation of the flow properties in respect to that the direct effects of emissions are masked and the zones of recirculating or stagnant air are rendered visible by local maxima in $\alpha$. Put in another sense, it could be also argued that regions marked by high values of $\alpha$ are especially prone to accumulation of air pollutants if additional emissions occur into that volume. A natural limitation of the proposed approach is clearly that $\alpha$ still depends indirectly on the distribution of emissions. However, as seen in this simplified example, the overall spatial patterns can nevertheless be expected to be rather insensitive in this regard.

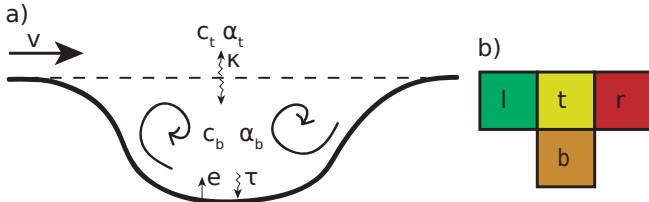

**Figure 1.** (a): Schematic overview of the different processes influencing the average concentration $c_b$ and age concentration $\alpha_b$ within an orographic basin. The processes include horizontal advection ($v$), vertical diffusion ($\kappa$), emission ($e$) and deposition ($\tau$). (b): A highly simplified spatial discretization to solve Eqs. 1-2 for the model depicted in (a).





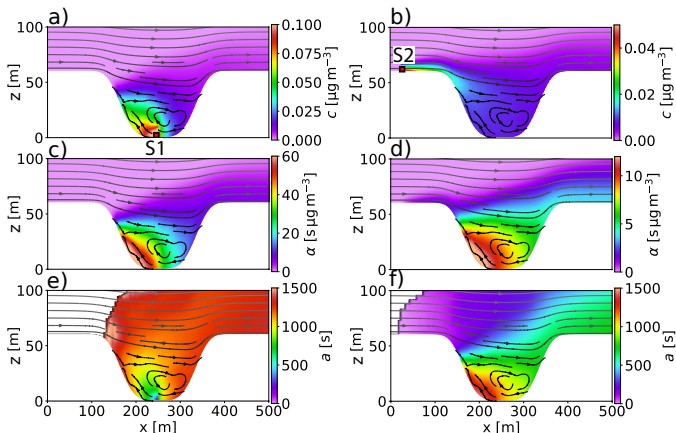

**Figure 2.** Numerical solutions of the concentration field $c$ (a, b), the age concentration field $\alpha$ (c, d), and the mean particle age field $a$ (e, f) for idealized 2-D (periodic in the y-direction) dispersion experiments. In experiment one (a, c, e), the emission source (red pixel, label "S1") is located at the bottom of the pot-shaped depression, in experiment two (b, d, f), (label "S2") it is located outside the depression near the left domain boundary.

## 2.2 Real-case simulation study

### 2.2.1 Model description

The urban large-eddy simulation model CAIRDIO is suitable to simulate emission, transport, and deposition of chemically inert air pollutants over complex topography at spatial resolutions ranging from the lower end of the mesoscale down to the
microscale. Two different approaches are used to represent topography in the model. The surface orography, which excludes buildings, is represented using terrain-following coordinates, while the buildings themselves are physical obstacles represented by diffuse-obstacle boundaries (DOB) in the Cartesian framework. CAIRDIO solves the incompressible fluid-dynamics equations with the anelastic approximation and employs a simple TKE-based subgrid model of the Smagorinsky type (Deardorff, 1973) to consider the effects of unresolved turbulent motions. Surface fluxes of momentum, heat and moisture are parameter-
ized based on Monin–Obukhov similarity theory, whereby surface temperature and surface humidity are provided by a special downscaling scheme from a mesoscale host model. More details on this downscaling scheme can be found in Appendix C1. In addition to the default transport equation Eq. 1 for air pollutants, Eq. 2 was implemented in CAIRDIO to adapt the model to the required capabilities in the framework of this study. For the solution of the resulting time-discrete coupled system of equations, it is first advanced Eq. 1 in time with a suitable explicit scheme. The arithmetic average of the old and new states
of $c$ is then used as the aging term in Eq. 2. The transport terms in both equations are discretized with a flux-limited linear advection scheme providing positivity preserving solutions for $c$ and $\alpha$. It must be noted, however, that for the mean-particle age $a$ computed as the quotient of $\alpha$ and $c$, positivity of the solution is generally not guaranteed unless a more strict flux-limiting criterion is used (Deleersnijder et al., 2020).





### 2.2.2 Model setup

The simulation domain (labeled D4), to which CAIRDIO is applied, covers an area of roughly $29\,\mathrm{km}$ x $25\,\mathrm{km}$ with $60\,\mathrm{m}$ horizontal resolution (490 x 420 grid cells in the horizontal plane). The domain is centred around the city of Dresden and incorporates the entire cross-section of the Elbe valley section with its surrounding elevated planes. (see Fig. 4 for an overview). The vertical grid encompasses 32 layers, with the first model layer following the surface orography, which is inferred from elevation data (DGM1) provided by the Staatsbetrieb für Geobasisinformation und Vermessung Sachsen (GeoSN). The surface

orography is progressively smoothed out with increasing height, and the upper-most model layer has a constant altitude of $1240\,\mathrm{m}$ above sea level. A moderate grid stretching is applied in the vertical direction, leading to a vertical resolution of $5\,\mathrm{m}$ near the ground and $120\,\mathrm{m}$ near the domain top. To represent buildings with DOB, effective grid-cell volumes and cell-face areas are computed using data from the Saxony-wide building-geometry model (level of detail LOD1), which is also available from GeoSN.

Realistic BC emissions for industry, housing, and transport are based on the Selected Nomenclature for Air Pollution (SNAP) and provided by the German Environmental Agency (UBA). While the road-transport emissions are represented by line sources, all other emissions are originally represented by area sources with $500\,\mathrm{m} \times 500\,\mathrm{m}$ resolution. To match our high-resolution requirements with domain D4, the area sources are further aggregated to either the sites of residential or commercial buildings, depending on the emission category. For the emission redistribution, respective building volumes are used to compute weight-

ing factors. For the city area of Dresden, spatially more accurate road-transport line emissions for primary particulate matter (PPM10) are provided by the Sächsisches Landesamt für Umwelt, Landwirtschaft und Geologie (LfULG). A BC/PPM10 conversion factor was derived using the UBA data to derive BC emission rates for this latter dataset (see Fig. 5 for the merged BC emissions within the first model layer). The static emissions are modulated in time using a time profile from the MACC-TNO database (Kuenen et al., 2014), which considers monthly, daily, and hourly variations in the activity of the different SNAP

categories.

Realistic meteorological and atmospheric composition boundary conditions with $180\,\mathrm{s}$ time resolution are provided with precursor simulations using the coupled weather prediction and air-chemistry transport model system COSMO-MUSCAT (Wolke et al., 2012). COSMO-MUSCAT is applied to a hierarchy of domains with increasing spatial resolutions using a one-way nesting approach. The configuration starts with an outer domain D0, which covers Germany and other parts of Central Europe with

a horizontal resolution of roughly $7\,\mathrm{km}$ (Fig. 3). For initialization of the regional model and as lateral boundary conditions, 3-hourly re-analysis data of the operational weather prediction model ICON-EU run by the German Weather Service (Deutscher Wetterdienst, DWD) are used. For the air-chemistry part, the BC transport simulation is driven by data from a European-wide COSMO-MUSCAT simulation with $14\,\mathrm{km}$ resolution. This latter simulation used re-analysis data from the operational ICON global model, and air-chemistry data from the ECMWF IFS model (Copernicus Atmosphere Monitoring Service) (Flemming

et al., 2015). Domains D1, D2, and D3 have horizontal resolutions of $2.8\,\mathrm{km}$, $1.4\,\mathrm{km}$, and $0.7\,\mathrm{km}$, respectively. For domain D3, the double-canyon effect parameterization DCEP (Schubert et al., 2012) is applied to simulate the effects of buildings and impervious ground surfaces on the mesoscale atmosphere, which, e.g., also includes the urban heat island effect. The prognos-





tic fields of 3-D wind, air temperature, specific humidity, cloud- and rainwater content, and TKE from domain D3 are spatially
interpolated to the CAIRDIO domain D4 using trilinear interpolation. Furthermore, computed urban surface temperatures of
building walls and roofs are diagnostically applied in the model CAIRDIO as the temperatures of respective urban surfaces
there. For the temperature and specific humidity of non-urban ground surfaces, as well as for surface pressure, which is needed
to reconstruct a hydrostatically balanced reference state, a special down-scaling method based on multiple linear regression is
applied using respective 2-D fields of mesoscale simulation D3. This method is explained in more detail in Appendix C1.

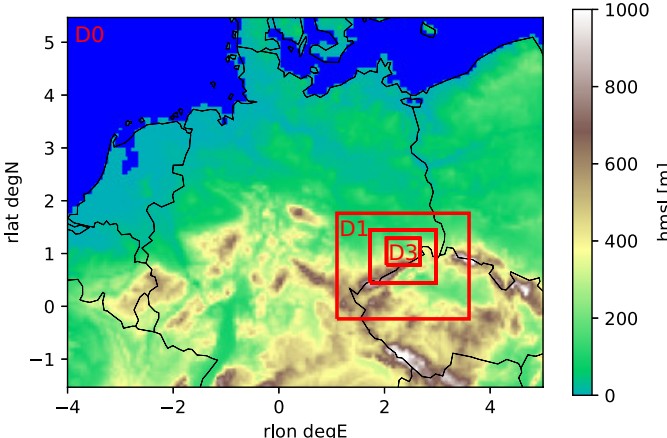

**Figure 3.** Overview of the different nested COSMO-MUSCAT domains indicated by red boxes to downscale meteorology and air composi-
tion towards the city area of Dresden. Domain D0 has a horizontal resolution of approximately 7 km, D1 2.8 km, D2 (not labeled) 1.4 km,
and D3 0.7 km.

### 2.2.3   Dispersion experiments

CAIRDIO is applied to simulate the boundary-layer evolution, as well as to host multiple BC dispersion experiments for a
total time span of 24 days. This duration encompasses two separate simulation periods, each one 12 days long. Period T1 is
in late winter 2021 from 27 February, 00:00 UTC to 11 March, 00:00 UTC, and period T2 in early summer 2021 from 31
May, 00:00 UTC to 12 June, 00:00 UTC. As for the BC transport simulation part, a first dispersion experiment is performed
that uses the emissions described in Section 2.2.2 and lateral boundary conditions of the precursor air-quality simulation D3
with COSMO-MUSCAT. This experiment (labeled BC-full) has the purpose to include a model validation in our study by
comparison of model results with BC measurements from operational air-monitoring stations. The results of this validation
are presented in Appendix A. The actual dispersion experiment relevant to the study of orographic effects uses the same BC
emissions as experiment BC-full, but solves the system of Eqs. 1-2. In difference to BC-full, no lateral boundary conditions
are prescribed for the mass and age concentration (i.e. homogeneous zero Dirichlet conditions are used). This approach is
equivalent to a tagging of the local emissions, for which reason, this simulation run is labeled BC-tag. Note that the regional





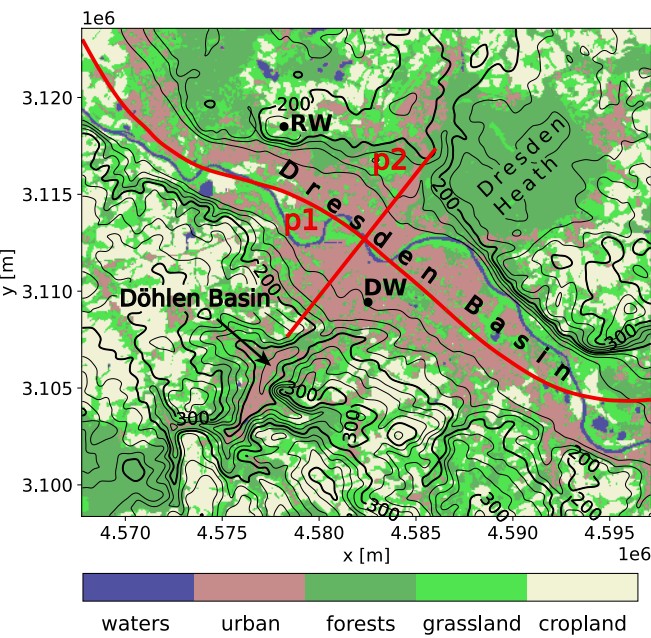

**Figure 4.** Overview of simulation domain D4 simulated with CAIRDIO. Shown are the distribution of land use, smoothed surface orography by black contours with $25\,\mathrm{m}$ spacing, the position and label of the background air monitoring stations Dresden Winckelmannstr. (DW) and Dresden Radebeul (RW) (see Section A for a description of the sites) by black dots and labels, and the paths p1 and p2 followed to extract vertical profiles of model data by red lines and labels.

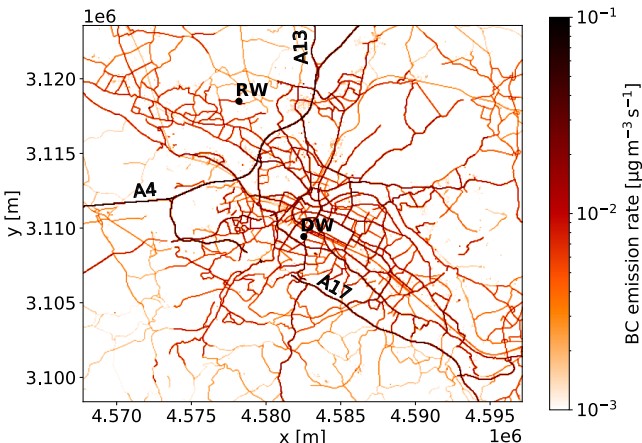

**Figure 5.** Distribution of BC emissions within the first vertical layer of domain D4. Elevated emissions from industrial sites and residential buildings are not shown. The positions of the background air-monitoring stations DW and RW are indicated by black dots and labels.





background contribution, which is missing in BC-tag, is expected to be anyways spatially homogeneous and only time variable at the local scale. In a third and final experiment, the realistic emissions are replaced by horizontally homogeneous emissions with a constant in space and time emission rate per horizontal unit area, while all other model settings apply to experiment BC-

tag. This experiment is labeled BC-hom and used to test the sensitivity of the age concentration distribution to the emissions.

## 3    Results

### 3.1    Meteorological conditions

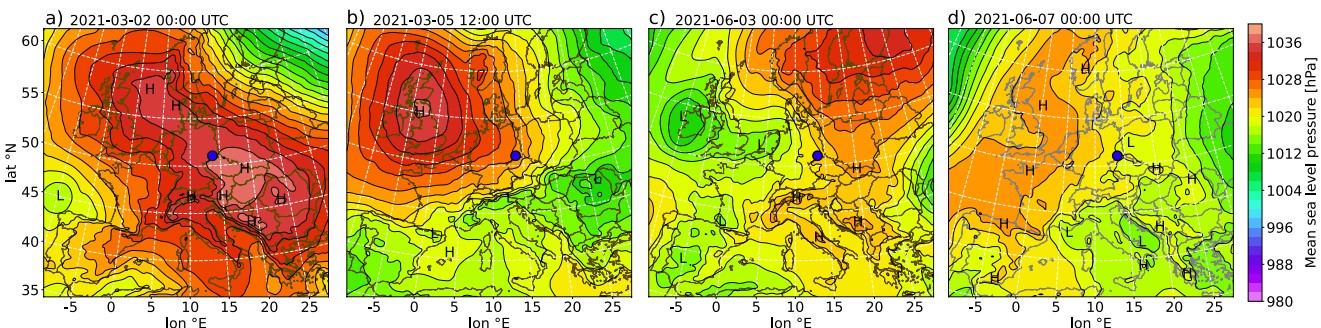

**Figure 6.** Synoptic-scale maps of mean sea level pressure over parts of Europe as considered representative for the prevailing weather patterns during simulation period T1 (a, b) and T2 (c, d), respectively. The maps are based on re-analysis data of the global forecast model ICON (13 km) of the DWD. The blue filled circle marks the geographical position of the city of Dresden.

### 3.1.1    Period T1: 27 February - 10 March 2021

The synoptic weather situation during the first simulation period T1 ranging from 27 February 2021, 00:00 UTC to 11 March

2021, 00:00 UTC was dominated by a blocking high-pressure system over northwestern Europe, which persisted from the beginning of the simulation period until 9 March and also temporarily expanded over central and southeastern Europe, so as from 28 February to 3 March (see Fig. 6a). Westward shifts of the high over the British Isles prompted the intrusion of cold continental airmass over eastern and central Europe, with one significant such event occurring on 4-5 March (Fig. 6b), and one less pronounced also on 8 March. Near the end of the simulation period, the blocking pattern over Europe resolved

as northwestern Europe was increasingly affected by an Icelandic low (not shown). The implications of the synoptic-scale meteorological situation on the temporal evolution of the boundary-layer structure in the Dresden Basin are shown in Fig. 7a-c. Fig. 7a shows the prevailing horizontal wind components $u$ and $v$ above the basin's average sidewall height ($> 300\,\mathrm{m}$), while Fig. 7b shows the along-valley wind speed within the basin. The first two simulation days were still influenced by northwesterly winds (generally below $7\,\mathrm{m\,s^{-1}}$), which translated to an up-valley (west-northwest) wind within the basin. The up-valley wind

was the strongest during daytime (up to $6\,\mathrm{m\,s^{-1}}$), presumably as a result of convectively enhanced vertical momentum transport

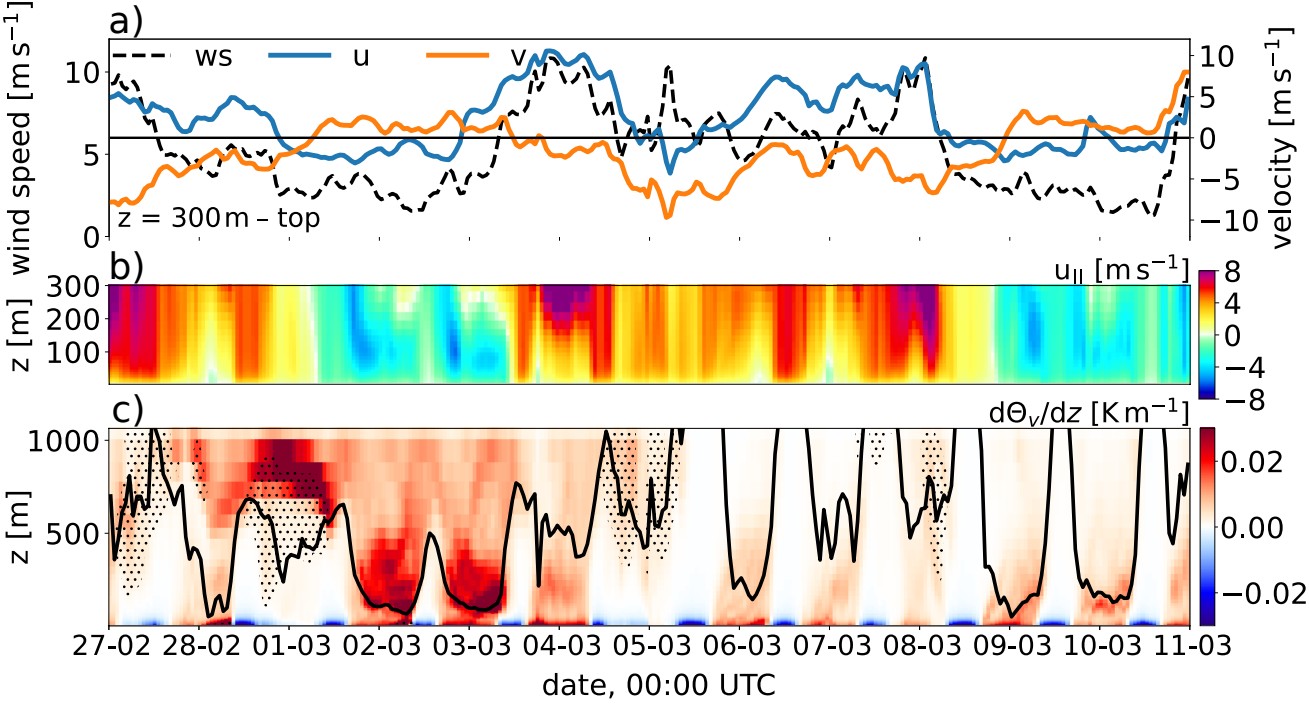

**Figure 7.** Model data averaged along path p1 (shown in Fig. 4) vs. time for the late winter simulation period T1: In (a) the wind velocity components $u$ (blue line) and $v$ (orange line), and the resulting wind speed (black dotted line) are averaged for the height range from $300\,\text{m}$ to the domain top. (b) shows the vertical profile of the along-valley wind speed, which is defined by the projection of the horizontal wind vector on the local tangent vector to p1. Positive values are related to up-valley winds. (c) shows the vertical gradient of virtual potential temperature (color plot), the boundary-layer height estimated from the bulk Richardson number (black line) and cloudiness (dotted pattern).

(see Fig. 7c for the stratification and boundary-layer height). As the high pressure expanded over central Europe over the next consecutive days 2-3 March, the prevailing winds shifted to the southeast and further weakened to below $4\,\text{m}\,\text{s}^{-1}$. This led to the formation of down-valley (east-southeasterly) winds, which were this time the strongest during nighttime. In particular, a nocturnal low-level jet can be identified by distinct wind maxima in about $100\,\text{m}$ height (wind speed up to $5\,\text{m}\,\text{s}^{-1}$). The jet is presumably a result of CAP formation within the Dresden Basin (also evidenced by the elevated maxima in the vertical gradient of $\Theta_v$) and the resulting vertical decoupling of the basin atmosphere and the free troposphere aloft. Note that this decoupling is also evident by the picking-up westerly winds after 3 March, 00:00 UTC, which do not mix down into the basin atmosphere until after sunrise. During the course of 3 March, westerly winds further increased to up to $10\,\text{m}\,\text{s}^{-1}$ and eventually turned to the north with the aforementioned intrusion of cold air on 4 March. The quite strong winds prevented the nocturnal stabilization of the boundary layer so that up-valley winds consistently prevailed from 4 March 00:00 UTC to 6 March 00:00 UTC. Thereafter, the wind speed temporarily decreased to allow for intermittent nocturnal decoupling of the basin atmosphere (e.g. in the early morning of 6 June and on June 7 around 00:00 UTC). Northwesterly winds briefly increased again during the night from 8-9





March. Thereafter, a period characterized by weak southeasterly winds aloft and down-valley winds within the basin with an apparent nocturnal jet resumed until near the end of the simulation period.

### 3.1.2 Period T2: 31 March - 11 June 2021

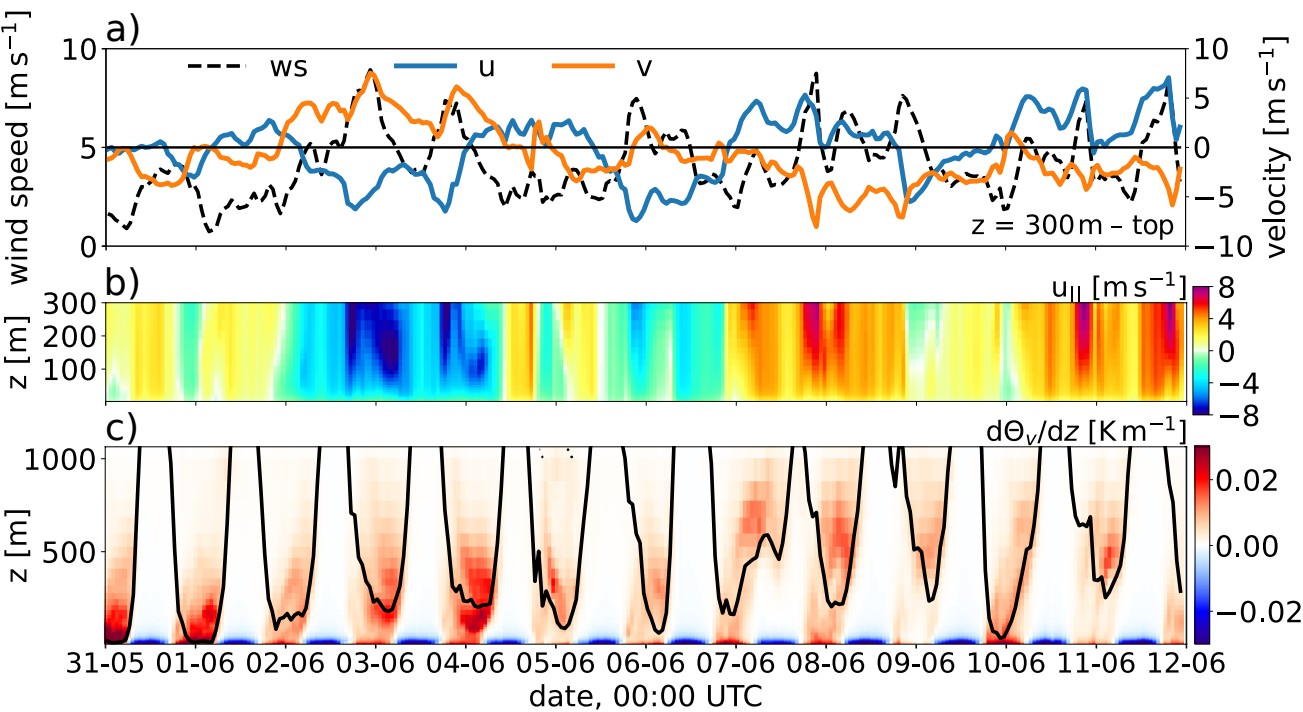

**Figure 8.** Same as Fig. 7, but for the early summer simulation period T2.

Similar to simulation period T1, a blocking pattern over Europe characterized also simulation period T2. The core of high surface pressure was, however, this time located over Scandinavia and western Siberia instead of northwestern Europe. This pattern allowed Atlantic disturbances to protrude more easily towards western and central Europe during the first half of the simulation period (Fig. 6c). On 4-5 June, an eastward moving cut-off low was associated with the development of locally intense thunderstorms over parts of France, Benelux, and Germany (not shown). The easternmost regions of Germany were, however, only marginally affected by this activity. On 6 June, high pressure built back over western Europe to form an extensive high-pressure ridge stretching from southwestern Europe all the way to Scandinavia (Fig. 6d). Central Europe remained under the influence of this high-pressure ridge until the end of the simulation period. Figures 8a-c show again the evolution of the PBL structure in the Dresden Basin during this time period. During the first two simulation days, generally low-wind conditions with weak along-valley winds prevailed (below $5\,\mathrm{m\,s^{-1}}$). During the nighttime, the basin atmosphere became stably stratified and a ground-based inversion layer (seen by the large $\Theta_v$-gradient near the surface in Fig. 8c) could form due to weak to absent





along-valley winds. On 2 June, southeasterly winds ahead of an approaching trough increased to up to $8\,\mathrm{m\,s^{-1}}$. As a result, also the wind direction within the basin shifted and a significant down-valley wind developed. In fact, the wind maxima between $200\,\mathrm{m}$ and $300\,\mathrm{m}$ height (up to $8\,\mathrm{m\,s^{-1}}$) can be associated again with the nocturnal low-level jet, which also coincides with the elevated maxima in the $\Theta_v$ gradient (Fig. 8c). Note, however, that despite the similarities with days 2-3 March of period T1, the thermal stratification is less stably stratified this time (cf. Fig. 7a-c). This is arguably due to a cyclonic influence over Germany at this time (Fig. 6c). As a weakening low moved eastward across France and southern Germany over the following two days 4-5 June, the southeasterly winds diminished and the along-valley winds shifted to a pattern of weak up-valley winds during daytime and weak down-valley winds during nighttime. From 7 June on, the area was situated at the eastern flank of the aforementioned high-pressure ridge to the northwest with the wind direction shifting to northwesterly to northeasterly winds. Close to the surface, the winds turned more to the west from the effect of surface friction, with the winds in the Dresden Basin blowing in the up-valley direction for most of the time.

## 3.2 Temporal mean BC dispersion patterns

For a first overview of the dispersion experiments, the temporal mean horizontal distributions of the BC concentration within the first model layer are discussed. In Fig. 9 respective patterns are shown for experiments BC-tag and BC-hom for the late winter period T1 and the early summer period T2, respectively. The patterns of experiment BC-full are not shown, as they differ only by a spatially near-uniform background offset of $0.28\,\mathrm{\mu g\,m^{-3}}$ for period T1 and $0.23\,\mathrm{\mu g\,m^{-3}}$ for period T2 from BC-tag. Firstly, it can be noted that the concentration distributions of experiment BC-tag (Fig. 9a, c) are strongly dominated by the traffic emissions, especially from the highway A4, which crosses the Dresden Basin in a northeasterly direction, but also from other major roads within the city area of Dresden. Aside from these prominent line features which exceed a concentration of $1\,\mathrm{\mu g\,m^{-3}}$, accumulations of BC with a spatially more uniform distribution can be discerned within the Dresden Basin. This is especially the case during the generally more stable period T1 (Fig. 9a) when average concentrations amount to at least $0.3\,\mathrm{\mu g\,m^{-3}}$. In this regard, the sharp gradient along the northern fringe of the basin is also striking, which is rendered visible by the underlaid terrain shading. The average concentration level during period T1 is higher than during period T2 ($0.18\,\mathrm{\mu g\,m^{-3}}$ vs. $0.10\,\mathrm{\mu g\,m^{-3}}$). This difference cannot be explained by a different emission rate, which on average is $1.48\,\mathrm{kg\,h^{-1}}$ over the entire domain D4 volume during period T1 and $1.35\,\mathrm{kg\,h^{-1}}$ during period T2. We also briefly discuss the BC dispersion patterns of experiment BC-hom at this point, which are further used in the subsequent paragraph of Section 3.4. In contrast to BC-tag, in experiment BC-hom, the BC distributions are solely determined by meteorology. As a result of the spatially homogeneous emissions, the bulk of BC mass is distributed over a wider area compared to experiment BC-tag, where it is more focused over the city area. The effect of surface orography is very eminent, as higher-than-average concentrations are generally found within topographic depressions, like the Dresden Basin, which sticks out during both simulation periods. During period T1 (Fig. 9b), a marked variability within the Dresden Basin is also apparent, as prominent maxima of roughly $0.3\,\mathrm{\mu g\,m^{-3}}$ are present to the north and south of the Dresden city area and over the southern basin side wall. Within the remaining parts of the Dresden Basin, concentrations average out at $0.15\,\mathrm{\mu g\,m^{-3}}$. Aside from the Dresden Basin, high concentrations are also reached within the southern tributary valleys. The most striking feature here is the Döhlen Basin containing the city of





Freital, wherein the domain-maximum concentration of about $0.5\,\mu\mathrm{g\,m^{-3}}$ is reached. For period T2 (Fig. 9d), the focus of BC concentration maxima is shifted from the southern tributary valleys to some shallow depressions north of the Dresden Basin, where concentrations reach up to $0.2\,\mu\mathrm{g\,m^{-3}}$ (e.g. over parts of the Dresden Heath, a large forest area to the northeast of the city).

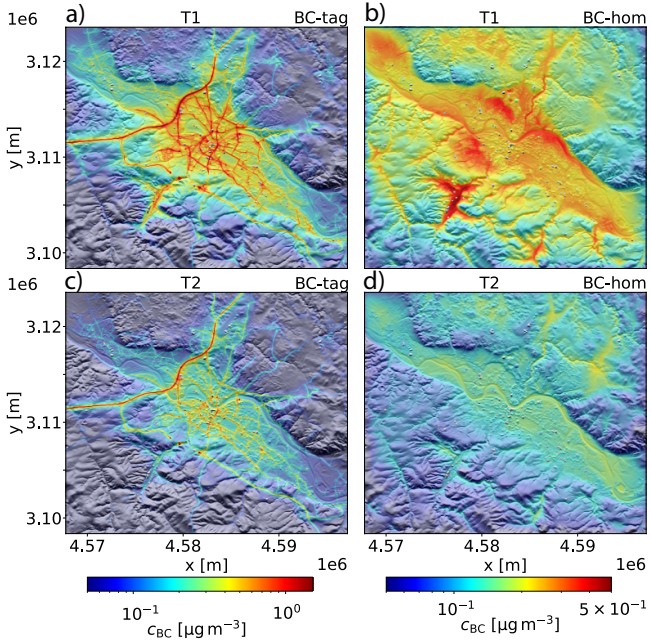

**Figure 9.** Temporal mean BC concentration $c_{\mathrm{BC}}$ patterns of dispersion experiments BC-tag (a, c) and BC-hom (b, d). The concentration in BC-hom is rescaled such that the area-sum equals that of experiment BC-tag. Note that different color scales are used for BC-tag and BC-hom. The patterns (a, b) refer to the simulation period T1, the patterns (c, d) to period T2.

### 3.3 Age-concentration modeling - proof of concept

### 3.4 Age-concentration modeling - proof of concept

While the BC concentration patterns of experiment BC-tag provide an overview of the simulated air pollution situation and also show the contrast between the late winter and early summer period, it is not yet conclusive which orographic features act as secondary pollution sources under the most realistic scenario studied. On one hand, the use of realistic emission data results in intrinsic spatial inhomogeneities that are superimposed on the spatial features induced by orography, making it potentially difficult to disentangle both effects. On the other hand, the simulation using spatially homogeneous emissions provides only an overview of potential orographic hot spots, but from this perspective, it cannot be concluded if an orographic region with significant air pollutant accumulations is also relevant when imposing a more realistic emission scenario. At this point, the age-concentration concept, which was introduced in Section 2.1.2 in the framework of air-pollutant dispersion modeling, may





provide a more conclusive picture in this regard. Similar to Fig. 9, Fig. 10 shows the temporal mean BC dispersion patterns for
       experiments BC-tag and BC-hom, but now for the age concentration $\alpha_{\mathrm{BC}}$ instead of the mass concentration $c_{\mathrm{BC}}$. As already
       pointed out, the direct effects of the emissions are masked in the age-concentration fields, and the regions that are characterized
       by stagnant or recirculating air mass are highlighted by high values of $\alpha_{\mathrm{BC}}$. The $\alpha_{\mathrm{BC}}$ patterns of the dispersion experiment
       BC-tag show that the entire modeled section of the Dresden Basin is very well marked-off by elevated values compared to the

adjacent higher-elevated planes, especially along the northern fringes of the basin. The intra-basin distributions of $\alpha_{\mathrm{BC}}$ show
       some marked differences between simulation periods T1 and T2. During period T1 (Fig. 10a), the focus is more on parts of the
       city area ($\alpha_{\mathrm{BC}} \approx 1.2 \times 10^3\,\mathrm{s\,\mu g\,m^{-3}}$) but even more so over the southwestern sidewalls of the basin ($\alpha_{\mathrm{BC}} \approx 1.5 \times 10^3\,\mathrm{s\,\mu g\,m^{-3}}$)
       and the Döhlen side basin, wherein $\alpha_{\mathrm{BC}}$ exceeds $2.5 \times 10^3\,\mathrm{s\,\mu g\,m^{-3}}$. During simulation period T2 (Fig. 10c), $\alpha_{\mathrm{BC}}$ is much
       more symmetrically distributed with generally lower values compared to period T2. The highest values occur over the city

center ($\alpha_{\mathrm{BC}} \approx 10^3\,\mathrm{s\,\mu g\,m^{-3}}$). Note that the lower values towards the domain borders are likely related to the zero-Dirichlet
       boundary conditions prescribed for $c_{\mathrm{BC}}$ and $\alpha_{\mathrm{BC}}$ there. Comparing the patterns of experiment BC-tag with experiment BC-
       hom (Fig. 10b, d) shows that the most important orographic hot spots identified by $\alpha_{\mathrm{BC}}$ in BC-tag are also present in BC-hom.
       For example, quite similar values of $\alpha_{\mathrm{BC}}$ are reached within the Döhlen Basin in both experiments, and $\alpha_{\mathrm{BC}}$ is also elevated
       over the southwestern side walls of the Dresden Basin in experiment BC-hom during period T1. Over the central parts of the

Dresden Basin, where the city center is located, $\alpha_{\mathrm{BC}}$ is lower by about $20\,\%$ in experiment BC-hom compared to BC-tag. This
       difference stems from the much more concentrated emissions over the urban areas in BC-tag compared to BC-hom. On the
       other hand, there are also $\alpha_{\mathrm{BC}}$ accumulations apparent over some areas north of the Dresden Basin in Fig. 10d for experiment
       BC-hom and period T2, which are not seen in Fig. 10c. As already pointed out in the discussion of respective $c_{\mathrm{BC}}$ patterns,
       these potential areas are not active in the realistic-emission setup, as emissions are in fact only sparsely distributed in these

areas, like, e.g., the Dresden Heath. Comparing Fig. 9b with Fig. 10b and also Fig. 9d with Fig. 10d reveals that many of the
       local maxima seen in $c_{\mathrm{BC}}$ are even better defined in $\alpha_{\mathrm{BC}}$. In this regard, the stationary solutions of $c_{st} = e/\kappa$ and $\alpha_{st} = e/\kappa^2$ of
       the simplified model derived in Section 2.1.2 can be recalled, which describe the potentiated spatial contrast in the field of $\alpha_{\mathrm{BC}}$
       compared to $c_{\mathrm{BC}}$. Finally, the comparison between Fig. 10a and b, as well as Fig. 10c and d, supports the conjecture that $\alpha_{\mathrm{BC}}$
       is rather insensitive to the emission distribution. In effect, from the aforementioned similarities, it can be argued that the age

concentration is a suitable metric to identify orography-related pollution hot spots, especially since it allows the incorporation
       of realistic emissions. Finally, for the sake of completeness, the mean particle age $a_{\mathrm{BC}}$ of experiment BC-tag is shown in Fig.
       11. Compared to the distribution of $c_{\mathrm{BC}}$, the emissions are now visible by lower than average values. Conversely, the highest
       values of $a_{\mathrm{BC}}$ typically occur in areas with low emissions, like, e.g., the Dresden Heath, where $a_{\mathrm{BC}}$ exceeds $1\,\mathrm{hour}$. Casually,
       the effect of the trees is a deceleration of the horizontal wind speed and in turn an increased residence time for horizontally

advected BC. However, the potential for air pollution trapping becomes only relevant if significant emissions occur upwind of
       the area. In comparison to the Dresden Heath, this is the case for the Döhlen Basin, where over the side slopes similar high
       values of $a_{\mathrm{BC}}$ occur during period T1. Note that in general, $a_{\mathrm{BC}}$ is significantly higher over the planes to the south of the Elb-
       river basin during period T1 compared to period T2, which is indicative of the different meteorological conditions prevailing
       on average between both simulation periods.





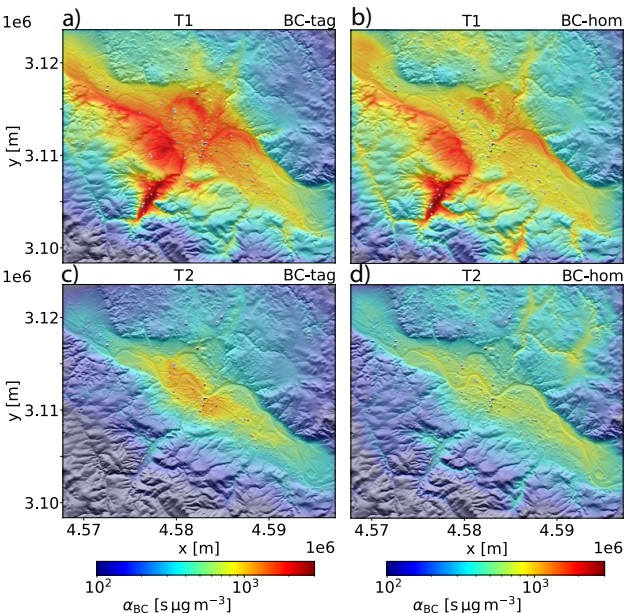

**Figure 10.** Temporal mean BC age-concentration $\alpha_{BC}$ patterns of dispersion experiments BC-tag (a, c) and BC-hom (b, d). The patterns (a, b) refer to the simulation period T1, the patterns (c, d) to period T2.

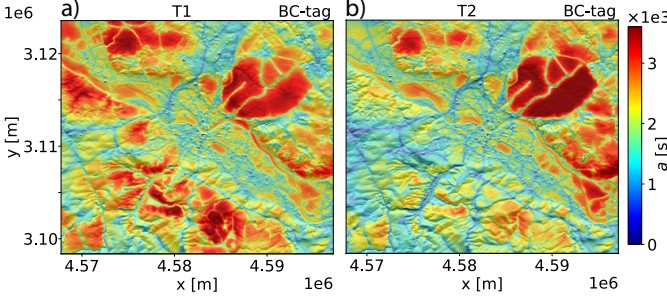

**Figure 11.** Temporal mean particle age $a_{BC}$ patterns of dispersion experiments BC-tag. The pattern (a) refers to the simulation period T1, the pattern (b) to period T2.

## 3.5 Principal-component analysis

## 3.6 Principal-component analysis

A principal-component (PC) analysis serves as a further step to incorporate also the temporally varying meteorological conditions in the analysis and at the same time to keep the complexity of the data low in terms of easy-to-interpret predominant atmospheric flow patterns. The PC analysis is applied on the near-surface wind field $v$, which are the horizontal velocity com-





ponents within the first model layer ($2.5\,\mathrm{m}$ height) of domain D4. In a first step, the temporal mean $\tilde{\boldsymbol{v}}$ is computed over both

simulation periods T1 and T2 together and then subtracted from the wind field to obtain the velocity fluctuations. The velocity

fluctuation field is then reshaped into a $2n_x n_y$-dimensional sample vector $\boldsymbol{v}'$, where $n_x$ and $n_y$ are the horizontal dimension

sizes of domain D4:

$$\boldsymbol{v}' = \left(u'_{11}, u'_{12}, u'_{13}, ..., u'_{n_x n_y}, v'_{11}, v'_{12}, v'_{13}, ..., v'_{n_x n_y}\right)^T \tag{17}$$

The PC analysis is performed by diagonalizing the covariance matrix $\mathbf{C}$ computed for $\boldsymbol{v}'$ (including both simulations periods

T1 and T2 together):

$$\boldsymbol{\Lambda} = \mathbf{E^T C E}, \tag{18}$$

where $\boldsymbol{\Lambda}$ is the resulting diagonal matrix containing the eigenvalues $\lambda_{1,2,3,...}$ in decreasing order. The columns $\boldsymbol{e}_i$ of matrix $\mathbf{E}$

are the estimated empirical orthogonal functions (EOF), i.e. they are orthonormal eigenvectors of $\mathbf{C}$. The eigenvalues indicate

the variance described by the associated PC $p_i$, which is the projection of the velocity fluctuation field on the corresponding

EOF ($p_i = <\boldsymbol{v}', \boldsymbol{e}_i>$). For a more robust capture of the physically relevant modes, it is often advised to resort to rotated EOF

analysis (Lian and Chen, 2012). The varimax rotation is applied on the first 50 conventional EOFs to obtain rotated EOFs.

Figures 12a and b show the streamlines of the first two rotated EOFs obtained for the combined period T1 and T2. The

corresponding PC series shown in Fig. 13 contain $82.5\,\%$ of the total variance of the data. EOF1 obviously describes the

variability of westerly and easterly winds while EOF2 describes the variability in the north-south direction. Hence, the first

two EOFs can be considered to form a base for the variability of the large-scale wind and the consecutive EOFs are more

related to noisy and insignificant small-scale variability ($f_{var} = 1.2\,\%$ for EOF3, not shown). It is also found that the first two

conventional not-rotated EOFs are approximately rotated in a $30\,^\circ$ angle with respect to the first two rotated EOFs (EOF1 west-

northwest to east-southeast aligned, EOF2 south-southwest to north-northeast aligned). So in fact, the most important effect

of the varimax rotation was just a rotation of the two horizontal base vectors in this case (apart from an increased contained

variance of PC2 by nearly $5\,\%$). The next step is to retrieve the corresponding prevailing wind patterns $\boldsymbol{v}_i^{\pm}$ for a negative

and positive amplitude of the first two EOFs, respectively. One possible approach is to use the associated PC in a temporal

weighting function, which can then be applied in the temporal-mean computation:

$$\boldsymbol{v}_i^{\pm} = \begin{cases} \frac{\sum_{k=1}^n \boldsymbol{v}(k)\max\left(0, p_i(k)\right)}{\sum_{k=1}^n \max\left(0, p_i(k)\right)} & \text{for positive PC amplitude} \\ \frac{\sum_{k=1}^n \boldsymbol{v}(k)\min\left(0, p_i(k)\right)}{\sum_{k=1}^n \min\left(0, p_i(k)\right)} & \text{for negative PC amplitude} \end{cases}, \tag{19}$$

where $n$ is the temporal sample size. Eq. 19 bears the advantage that it can be applied on any arbitrary model variable (not

only on the near-surface wind field) and also for each simulation period separately. In Fig. 12c and e, the results are shown for

the first EOF and both simulation periods T1 and T2 combined. Accordingly, a positive amplitude of the first EOF (EOF1+) is





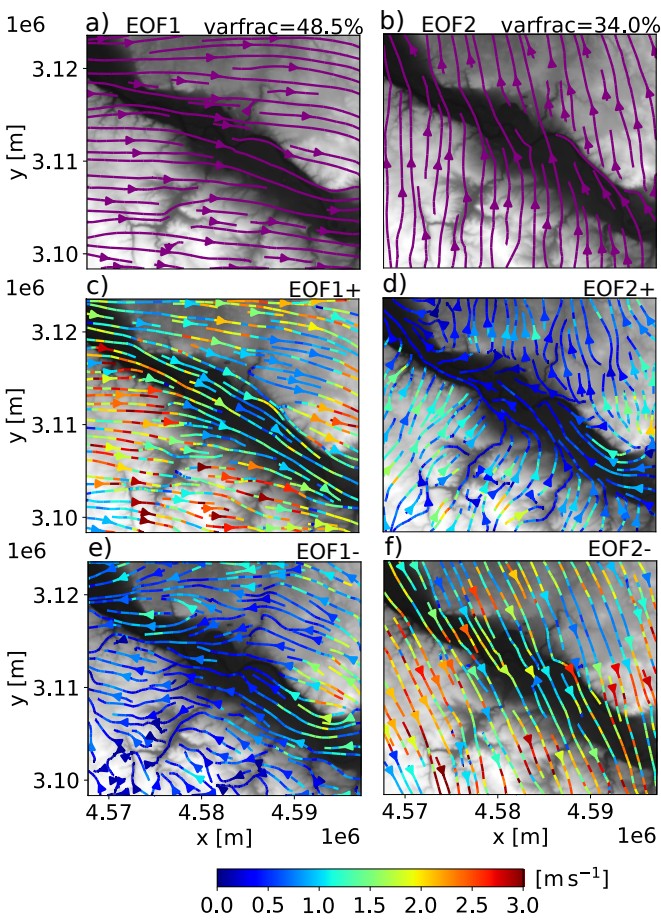

**Figure 12.** Streamline pattern of the first (a) and second (b) rotated EOF. $f_{var}$ indicates the percentage of the total variance described by the corresponding EOF. (c) and (e) show the average reconstructed near-surface wind field for a positive and negative amplitude (computed with Eq. 19 over period T1 and T2) of the first EOF, while (d) and (f) show the same information for the second EOF.

on average associated with west-northwesterly winds over the elevated planes (up to $3\,\mathrm{m\,s^{-1}}$) and an up-valley wind within the Dresden Basin. This pattern is associated with a generally weaker boundary-layer stratification and greater nocturnal mixed-layer heights (compare Fig. 13 with Fig. 8b and Fig. 9b). Conversely, for the case EOF1- (negative amplitude) the prevailing wind direction is from the east-southeast with peak winds of $2\,\mathrm{m\,s^{-1}}$ over the highest planes. This case also corresponds to a more stable boundary-layer stratification and the occurrence of the nocturnal low-level jet. As in both cases, the prevailing wind direction is relatively well aligned with the orientation of the basin, no large differences in the wind direction between the basin and planes occur. This is different for the case EOF2+ (Fig. 12d). Here, the prevailing wind direction is from the south-southwest over the planes south of the Dresden Basin and slightly rotated to the south-southeast over the northern planes, with generally lower wind speeds than in cases Fig. 12c and e (only up to $1.5\,\mathrm{m\,s^{-1}}$). Within the basin, a weak down-



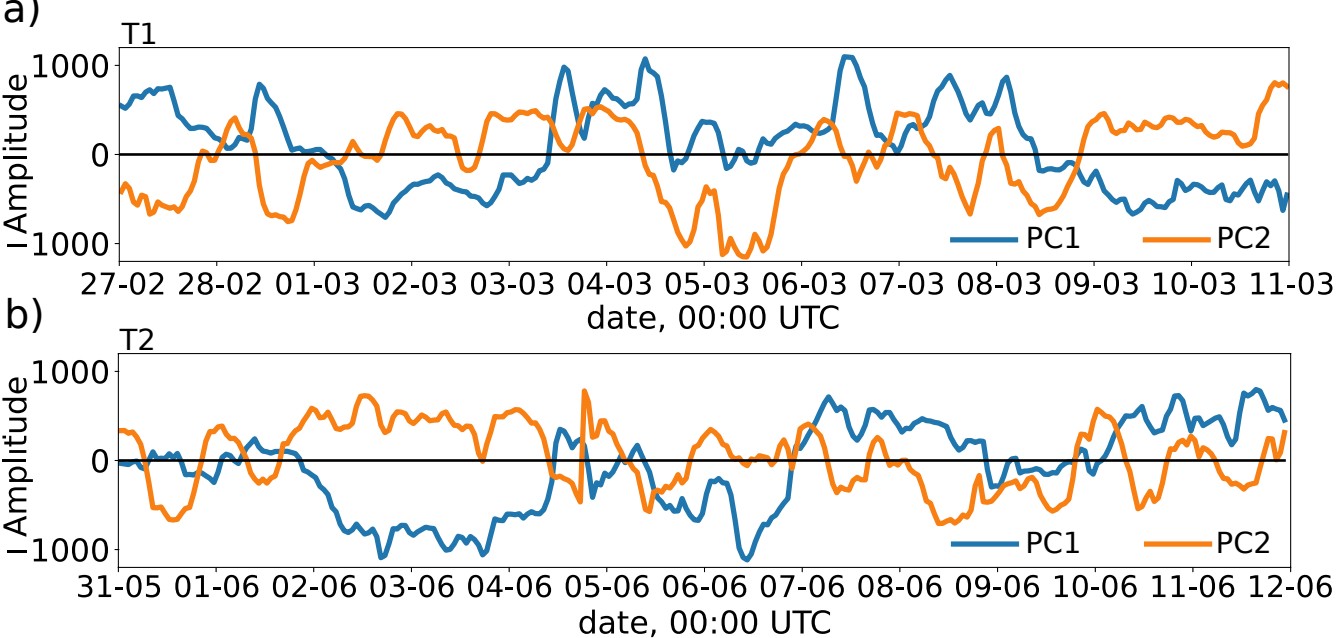

**Figure 13.** Principal-component (PC) time series PC1 and PC2 associated with the rotated EOF1 and EOF2 for simulation period T1 (a), and T2 (b), respectively.

valley wind (mostly below $1\,\mathrm{m\,s^{-1}}$) seems to be well aligned with the local orientation of the valley. Opposite to the EOF1, a positive amplitude of EOF2 (EOF2+) is apparently associated with a very stable stratification, which can be also verified when comparing again Fig. 13 with Fig. 8b and Fig. 9b. The fourth and final pattern (Fig. 12f) refers to the case EOF2-. This case

is associated with stronger northwesterly winds (up to $2.5\,\mathrm{m\,s^{-1}}$) and only a small contrast in both wind speed and direction between the Dresden Basin and the adjacent elevated planes. In fact, the streamlines run mostly straight across the domain with little apparent influence from the underlying topography, which is indicative of a significant turbulent boundary layer being present.

After the identification of the most important horizontal wind patterns, Eq. 19 is applied on the near-surface horizontal BC
concentration distribution of simulation run BC-tag. In contrast to the wind patterns, the two simulation periods are considered separately here. The resulting patterns are shown in Fig. 14. As can be seen, the different EOF patterns correspond also to quite a different magnitude in BC concentrations within the Dresden Basin. For example, cases EOF1- (Fig. 14b) and EOF2+ (Fig. 14c) show the highest concentrations during period T1. These two cases are also the only cases associated with markedly elevated BC concentrations within the Döhlen Basin ($c_{\mathrm{BC}} > 1\,\mathrm{\mu g\,m^{-3}}$). During simulation period T2, only case EOF2+ (Fig.
14g) gives evidence of air pollution trapping within the Dresden Basin. This is in contrast to case EOF2-, which shows the lowest average BC concentrations during both periods (Fig. 14d and h). To exclude that the observed differences are related to the emissions, the average emission rates are also computed with Eq. 19 and listed in Table 1. Accordingly, the emission rates





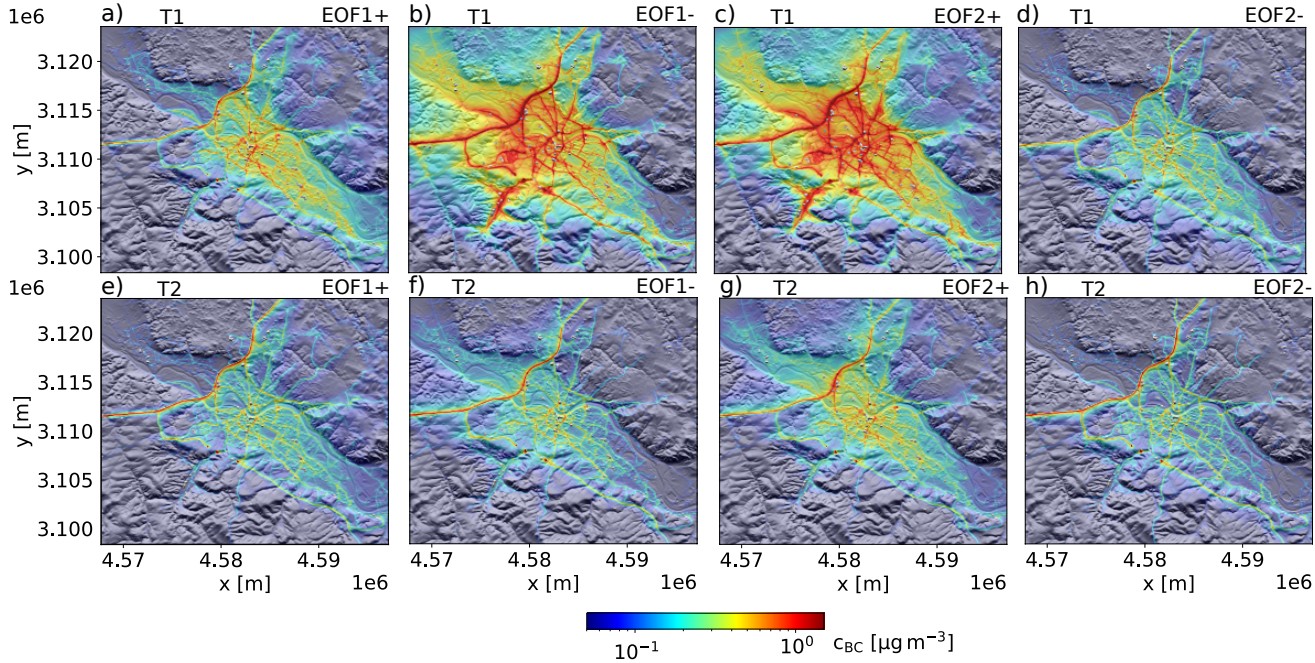

**Figure 14.** Patterns of the BC concentration within the first model layer computed with Eq. 19 for a positive and negative amplitude of the first (a,b,e,f) and second EOF (c,d,g,h), respectively. The patterns (a-d) refer to the late winter period T1, and the patterns (e-h) to the early summer period T2.

do not explain the observed concentration patterns, as for example, the emission rate differs only by $3\%$ at the most between the cases EOF2+ and EOF2-.

**Table 1.** Temporal mean BC emission rates computed with Eq. 19 and integrated over the domain D4 volume for the different EOFs and simulation periods. Units are in $\mathrm{kg\,h^{-1}}$.

| Periods | EOF1+ | EOF1- | EOF2+ | EOF2- |
|---------|-------|-------|-------|-------|
| T1 | 1.40 | 1.13 | 1.57 | 1.56 |
| T2 | 1.51 | 1.53 | 1.30 | 1.34 |

To corroborate the conjecture that the boundary-layer structure is indeed the determining factor in the observed differences, Fig. 15 shows vertical cross-sections of the BC concentration and mean particle age along path p2 (path shown in Fig. 4) for the two contrary cases EOF2+ and EOF2-. In addition, yellow contour lines show the stratification, and black contour lines the along-valley wind. The saturated blue colors indicate a high average particle age ($a_{\mathrm{BC}} > 1.5\,\mathrm{h}$), while pale blue colors indicate fresh aerosol. Similarly, saturated reds are associated with a high concentration $c_{\mathrm{BC}} > 0.5\,\mathrm{\mu g\,m^{-3}}$, and, finally, purple colors
mark areas where both a high particle age and high concentrations overlap. Several observations can be made based thereon.

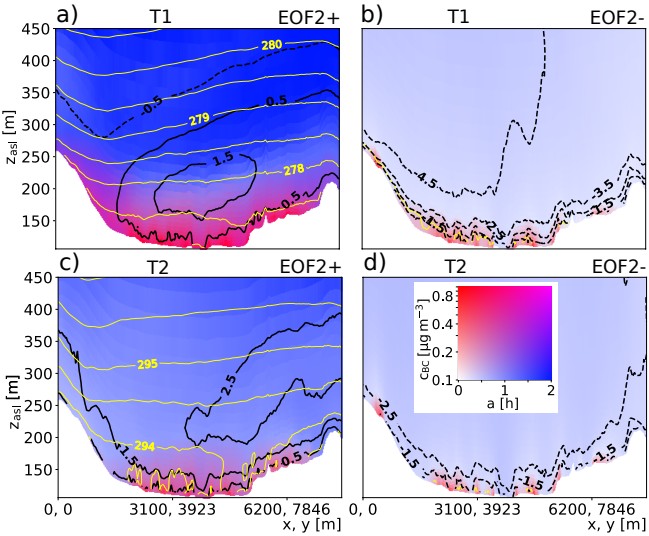

**Figure 15.** Vertical distribution of black-carbon mean-particle age $a$ (blue saturation, linear scale) and mass concentration $c_{BC}$ (red saturation, logarithmic scale) along the valley-perpendicular path p2 depicted in Fig. 4. Note that the vertical coordinate $z_{asl}$ refers to the height above sea level. The two different patterns for EOF2 are again computed with Eq. 19. In addition to the color plot, black contour lines show the magnitude and sign (dashed contours for negative sign) of the plane-perpendicular wind component (positive for the down-valley direction) and yellow contours show the virtual potential temperature ($0.5\,\mathrm{K}$ spacing).

Firstly, the EOF2+ case (Fig. 15a and c) is associated with a stably-stratified boundary layer (high density of yellow contour lines) compared to case EOF2- (Fig. 15b and d), which shows a near-neutral stratification. This coincides with a positive plane-perpendicular or down-valley wind component (solid black contour lines) for case EOF2+ and up-valley winds (dashed black contour lines) for case EOF2-. For the EOF2+ case, thermal stratification is more pronounced during period T1 (Fig. 15a) 495 compared to period T2 (Fig. 15c), and also the weak low-level jet core is better defined and confined to the basin interior during period T1 compared to T2. Secondly, a high mean particle age $a_{BC}$ in the upper model layers is correlated with a high mass concentration $c_{BC}$ near the bottom of the Dresden Basin and vise versa, which is irrespective of the simulation period. The case with the stable most stratification (EOF2+, T1) shows also the highest values in $a_{BC}$ and $c_{BC}$ near the surface among the cases considered. $a_{BC}$ reflects the large residence time of particles trapped within the Dresden Basin. Note in this regard that 500 the decrease in $a_{BC}$ towards the ground is an effect of the urban emissions (cf. Fig. 2e of the idealized experiment in Section 2.1.2). Finally, the overlapping regions of high $a_{BC}$ and $c_{BC}$ correspond also to the areas where elevated values of $\alpha_{BC}$ occur in the horizontal maps. In the case EOF2+ during period T2, these areas are mainly confined to the central parts of the basin over the city area. During period T1, however, the same areas appear in a more saturated purple and extend also significantly over the sidewalls of the basin, especially over the southern sidewall.



## 4    Conclusions

The urban dispersion model CAIRDIO was applied on the Dresden Basin, a widened section of the Elbe Valley that contains the mid-sized city of Dresden, to investigate the influence of surface orography on the spatial distribution of BC as an important air pollutant. Two separate periods in late winter (T1) and early summer (T2) 2021, each 12 days long and characterized by predominantly calm weather conditions, were simulated. Realistic meteorological and air composition boundary conditions, as well as surface forcing, were generated with a nested hierarchy of mesoscale simulations with the coupled meteorological and air-chemistry model COSMO-MUSCAT. Two different emission scenarios were considered. In scenario BC-tag, a realistic BC emission dataset considering traffic, industrial and residential emissions was used, which was also modulated in time to consider diurnal changes in activity. In a second scenario BC-hom, a spatially and temporally homogeneous BC emission rate was prescribed. Thus, in this dispersion experiment, the spatial distribution of BC was solely determined by the interaction of meteorology with topography. In addition to the simulation of emission, transport, and deposition of the mass concentration of BC, an additional similar transport equation for the BC age concentration was solved. As this transport equation does not directly depend on the emissions, it softens away the spatial inhomogeneities associated with the prescribed realistic emissions. Moreover, the age concentration typically reaches maximum values in areas characterized by stagnant or recirculating air, making it a useful metric to visualize air-pollution trapping within orographic depressions during stable weather conditions.

As expected, the temporal mean distributions of BC mass concentrations were dominated by the line features associated with the traffic emissions in scenario BC-tag. Simulated domain-average BC mass concentrations were nearly twice as high during period T1 compared to period T2. This difference could be attributed to a generally more stably-stratified boundary layer during period T1, though spatial differences related to surface orography remained inconclusive on basis of the mass concentration of scenario BC-tag alone. In contrast, the mass and age concentrations of scenario BC-hom, as well as the age concentration of scenario BC-tag showed a completely different picture that indeed highlighted orographic features and revealed also interesting differences between simulation period T1 and T2. During period T1, the Dresden Basin was marked off by higher values compared to the adjacent elevated planes. Maximum values were reached over the southwestern side slopes of the Dresden Basin, with the southern tributary Döhlen Basin appearing as the most prominent hot spot. During period T2, only the urban area within the central section of the Dresden Basin and some areas north of the Elbe Valley showed markedly elevated mass and age concentrations of BC. The main difference between the mass and age concentration of run BC-hom was that the age concentration field provided an even better differentiation of air-pollution trapping zones. This result can be explained by the simplified stationary solutions of the two respective transport equations. The similarities in the spatial age concentration distributions between scenarios BC-tag and BC-hom prove the robustness of the orographic flow information included in the property of the age concentration, as it is not very sensitive to the spatial distribution of emissions. Nevertheless, scenario BC-hom showed false indications over some areas where in reality no air pollutant emissions occur. The age concentration can therefore be considered a more appropriate metric that reflects the real distribution of air pollutant sources.

As a second result of the study, the most prevalent flow regimes during the two simulated periods were identified. Therefore, a statistical EOF analysis was carried out and the resulting patterns were further characterized based on their influence on local





air pollution exposure. The first two most important EOFs essentially represented two contrasting flow regimes: The first flow
regime could be characterized by a northwesterly prevailing wind direction and a stratification close to neutral. In contrast, a
southeasterly wind direction with generally weaker winds was associated with a much more stable boundary-layer stratification.
This resulted in the establishment of local orographic wind systems, like down-valley winds within the Elbe Valley. Only for
this second flow regime, a pronounced accumulation of air pollution within the Dresden Basin and its tributaries occurred.

This study provided a mainly phenomenological description of air pollution trapping within the Dresden Basin. An in-depth
analysis of the physical forcing mechanisms behind the observed spatial patterns was beyond the scope of this paper. This,
however, will be addressed in a future work using even higher-resolved domains targeting interesting sub-areas revealed by
this study, like, e.g. the Döhlen Basin.

*Code availability.* The model code of CAIRDIO v2.1, as well as utilities for data pre-processing used in this study are accessible in release
under the license GPL v3 and later at https://doi.org/10.5281/zenodo.7409604 (Weger et al., 2022b).

*Data availability.* All model results and observational data published in this study are accessible under the license GPL v3 and later at
https://doi.org/10.5281/zenodo.7410286.

**Appendix A: Model validation with ground-based in-situ measurements**

For a concise validation of the CAIRDIO simulations, meteorological (wind speed, wind direction, air temperature) and BC
observations are inferred from two air-monitoring sites operated by the LfULG. Station Dresden-Winckelmannstr. (labeled
DW) is located within the Dresden Basin in the southern part of Dresden, while station Radebeul-Wahnsdorf (labeled RW)
is located above the basin on a lift to the northwest of the city (see Fig. 4-5 for map overviews). Both stations are classified
as background stations since the closest major roads are more than $50\,\mathrm{m}$ distant from station DW (average daily traffic count,
ADTC of 27430 vehicles), and more than $100\,\mathrm{m}$ from station RW (ADTC of 4330 vehicles). Besides the different larger-scale
environments the stations are situated in (urban background within the basin for station DW, rural background above the basin
for station RW), the closer building environment is different for the two stations, as depicted by the sky-view simulations in Fig.
A1. Station DW is entirely surrounded by buildings, even though the average distance of most buildings from the measurement
site is above $50\,\mathrm{m}$. This results in a sky-view factor of $0.84$. The most significant obstruction is a roughly $25\,\mathrm{m}$ tall building
about $60\,\mathrm{m}$ to the northeast of the site. Significant tree cover is also present mainly to the east, south, and west of the site (not
shown). The wind measurements can thus not be regarded to be entirely representative of the prevailing wind conditions within
the Dresden Basin. In contrast to site DW, site RW provides a mostly unobstructed view of the sky (sky-view factor of $0.94$).
The only significant obstacle near the site is a lattice tower ($35\,\mathrm{m}$ height) located $15\,\mathrm{m}$ to the northwest of the station.

The comparisons of modeled wind speed with hourly wind speed measurements for the two simulation periods are shown in





Fig. A2a (period T1) and b (period T2). The measured wind speed in the Dresden Basin at site DW (grey lines) is generally low throughout the simulation periods as it fluctuates around $1 \, \mathrm{m \, s^{-1}}$. The fluctuations seem to temporarily follow a diurnal cycle with the lowest wind speeds (near calm conditions) generally occurring during the night-time hours. During period T1, the modeled wind speed (red lines) generally follows the measured profile well, as it reproduced the observed fluctuations and magnitude in wind speed. Nevertheless, the model overestimates the wind speed during some calm events during period T2, which suggests that the nocturnal boundary layer is not always accurately represented in the model. Without additional measurements to resolve the vertical structure of the boundary layer, this is, however, impossible to verify. At site RW located outside the Dresden Basin, the measured wind speed (black lines) is consistently higher than at site DW and reaches occasionally up to $5 \, \mathrm{m \, s^{-1}}$. Partly, this difference is also a result of the more unobstructed environment station RW is located within. The model (blue lines) again follows this trend well, even though there are also periods when the model significantly overestimates wind speed by at least $2 \, \mathrm{m \, s^{-1}}$ (1 March of period T1, 3-4 June of period T2). Concerning the validation of the wind direction, only the comparison for site RW is shown (Fig. A2c-d), as site DW is more influenced by buildings and would demand a higher resolved simulation for a representative comparison. The measured wind direction profiles at site RW express the variable synoptic influence already described in Sections 3.1.1 and 3.1.2. For example, they show the sequence of northwesterly and southeasterly winds, which the model generally follows quite well, especially during period T1. Significant disagreements occur on 31 May and on 5-6 June, when the model shows a more easterly wind direction instead of westerly to northwesterly winds. These disagreements are also seen in the mesoscale precursor runs with the model COSMO-MUSCAT (not shown). In Fig. A2e and f, measured profiles of the near-surface air temperature are compared with model results. Generally, the differences between the sites DW and RW are only marginal. Most of the time though, site DW shows higher temperatures than site RW, which is a result of the different elevations of the sites ($112 \, \mathrm{msl}$ vs. $246 \, \mathrm{msl}$). During a few nights, however, the air temperature at site DW dropped below the temperature at site RW, which indicates the presence of a diurnal CAP within the Dresden Basin (e.g. from 3 to 4 March). The model generally follows the measured temperature evolution well, especially in the longer term. For some individual diurnal cycles, however, the model deviates from the measurements by up to $5 \, \mathrm{K}$, and this is irrespective of the site. In particular, during some nights (3-4 March, 1-2 June, and 5-6 June) the modeled air temperature did not drop as low as the measurements show. Similar deviations are again seen in the mesoscale precursor simulations (not shown), which is not surprising given that a surface temperature interpolation scheme is used instead of a prognostic surface model. Lastly, Fig. A2g and h show the BC measurements and comparable model results of dispersion experiment BC-full (using realistic emissions plus realistic boundary conditions from the COSMO-MUSCAT simulation D3). During period T1, a clear relationship between a southeasterly to southerly wind direction and elevated BC concentrations at site DW (peaks above $2 \, \mathrm{\mu g \, m^{-3}}$) can be observed, which to a lesser extent applies also to site RW. The only notable exception is a marked peak (above $4 \, \mathrm{\mu g \, m^{-3}}$) at site DW during the night from 5-6 March, which is not present at site RW. During this time the prevailing wind direction outside the Dresden Basin was from the west, but due to the stable stratification calm wind conditions resulted within the basin at site DW. For the rest of the time, BC concentrations were mostly below $1 \, \mathrm{\mu g \, m^{-3}}$. Model results at site DW show a similar pattern, even though the aforementioned prominent peak from 5-6 March is missed by the model, probably as a result of an underestimated stability of the boundary layer during this time. Based on the computed fractional bias (FB) the model





significantly underestimates BC concentrations at both sites RW (FB = 0.75) and DW (FB = 0.39). Since the urban emissions and orographically induced air-pollution trapping effects have a larger influence on BC concentrations at site DW compared

to site RW, the better model agreement at site DW suggests that the regional BC background concentrations are significantly underestimated. This is confirmed by the comparison with the mesoscale model run D3, which shows a similar underestimation for site RW (not shown). Compared to period T1, the relationship between the prevailing wind direction and the measured BC concentration is much less clear during period T2, and the average BC concentrations (most of the time below $1\,\mu g\,m^{-3}$) are significantly lower compared to period T1. While the model still underestimates BC concentrations, the difference is this

time much smaller (FB = 0.20 at site RW and FB = 0.40 at site DW) but likely of the same cause as for period T1, i.e. an underestimated regional BC background.

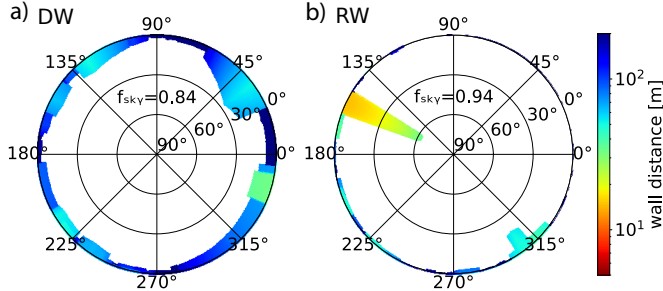

**Figure A1.** Simulated view on surrounding buildings in spherical coordinates at the exact 3-D locations of the inlets for the stations (a) DW, and (b) RW. The colors indicate the distance of instrument inlets to building walls, while the visible sky is shaded in white. Additionally, the sky-view factors $f_{sky}$ are computed as the fraction of the solid angle of the hemisphere not blocked by buildings.

## Appendix B: Simulation details of 2-D dispersion experiment in Section 2.1.2

The computation domain used in the idealized simulations presented in Section 2.1.2 is made up of $n_x \times n_y \times n_z = 120 \times 20 \times 60$ grid cells with a uniform spatial resolution of $5\,m$. The pot-shaped depression in the center of the domain is represented by

terrain-following coordinates with the terrain-height function given by

$$h_s(x) = \Delta h \left[ 1 - \exp\left( \frac{x - L/2}{0.15L} \right)^4 \right], \ \Delta h = 60\,m, \ L = 600\,m. \tag{B1}$$

The function $h_s(x)$ is progressively smoothed out along the z-axis, such that the domain-top boundary results in a flat surface. The domain is periodic in the y-direction. As for the meteorological initial conditions, a horizontally constant potential temperature profile is prescribed with a vertical gradient of $d\Theta/dz = 0.03\,K\,m^{-1}$. The potential temperature at the basin floor

($z = 0\,m$) is $280\,K$. A surface-roughness length of $z_0 = 0.03\,m$ and a horizontally uniform surface potential temperature of $\Theta_s = 280\,K$ are prescribed to model surface-heat fluxes. The specific humidity is set to zero for the atmosphere as for the



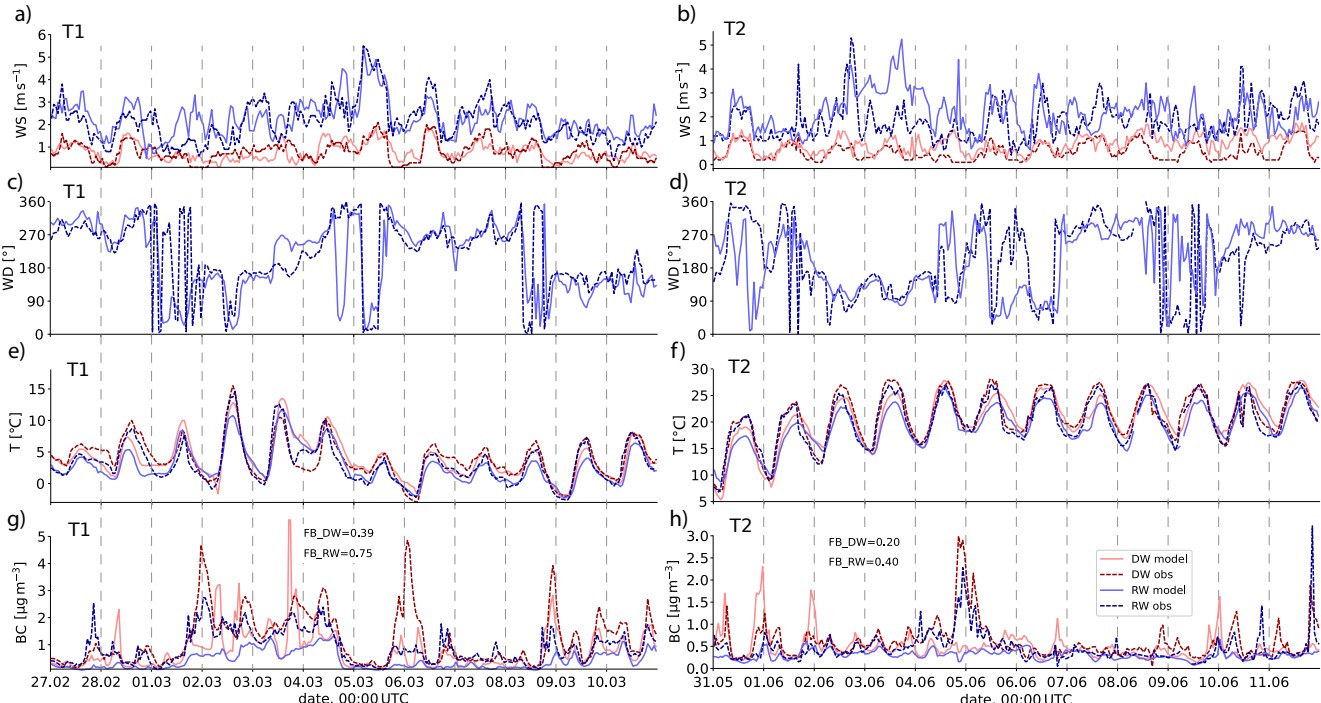

**Figure A2.** Model evaluation of run BC-full with meteorological and air-quality measurements from two background air-monitoring sites (site DW located within the Dresden Basin, site RW located on a lift to the north of the Dresden Basin, see Fig. 4). (a) and (b) show the comparison for the wind speed, (c) and (d) for wind direction (for site RW only), (e) and (f) for air temperature, and (g) and (h) for the BC concentration. For the BC concentration, the time-average fraction biases (FB) between measurements and model results are computed. Measurements from sites DW and RW are shown by dashed red and blue lines, respectively. Model results at the sites DW and RW are shown by full red and blue lines, respectively.

surface. The initial wind speed along the x-axis is $u = 5\,\mathrm{m\,s^{-1}}$, the same value is also continuously prescribed as the domain-top boundary condition for velocity. Using periodic boundary conditions in the x-direction for meteorology, a boundary layer develops with progressing simulation time, and simultaneously a CAP builds up within the sheltered basin. After a simulation period of $2\,\mathrm{h}$, only minor further changes to the boundary layer structure occur and the dispersion experiments can be conducted. Emission source S1 is placed near the basin floor at $x = 300\,\mathrm{m}$ and source S2 on the left basin rim at $x = 80\,\mathrm{m}$. Both sources extend infinitely along lines in the periodic y-direction and emit at a constant rate of $0.2\,\mathrm{\mu g\,m^{-1}\,s^{-1}}$ ($1\,\mathrm{\mu g\,s^{-1}}$ per grid cell).






## Appendix C: CAIRDIO v2.1 model updates

### C1 Improved diagnostic interpolation of surface variables

CAIRDIO depends on external surface fields for potential temperature $\Theta_S$, specific humidity $Qv_S$, and pressure $p_S$. While the former two fields are required in the computation of the surface fluxes for heat and moisture, $p_S$ provides the lower boundary condition in the derivation of a hydrostatically balanced reference atmosphere used in the anelastic approximation of the fluid-dynamic equations, and the cloud-saturation and radiative-transfer schemes. In our former version of CAIRDIO (v2.0), the surface fields were obtained by a diagnostic downscaling of respective prognostic data from the hosting COSMO

simulation (Weger et al., 2022a). The downscaling consisted of fitting the COSMO data into a linear regression model, which was based on land use as the only predictor. This approach proved to be computationally cheap and feasible, provided that surface orography was mostly flat. In the current model version v2.1 used in the framework of this study, the approach was extended to a more sophisticated multiple linear regression model to address this limitation. Land use is distinguished into 5 independent classes, namely forests, grassland, waters, bare soil, and urban areas. A spatially varying temperature field of urban

(impervious) surfaces is, however, already provided by the urban parameterization DCEP in COSMO and is simply interpolated using bilinear interpolation. Note that this particular approach is not followed for $Qv_S$ and $p_S$, where the urban surfaces are included in the regression model. Additionally to land cover, predictors in the revised scheme now include surface elevation $h_S$, surface normal orientation expressed by azimuth and elevation angles ($\phi_S, \psi_S$), cloud attenuation $\eta_{cl}$, precipitation rate $pr$, and 9 more coefficients $c_{ij}$ in a horizontal bicubic function $f(x,y) = \sum_{i=1}^{3}\sum_{j=0}^{i} c_{ij}x^j y^{i-j}$ to consider large-scale variations. The

importance of the different predictors clearly depends on the kind of variable to be fitted. For example, in contrast to $T_S$, $p_S$ will only negligibly depend on land use but predominantly on $h_S$. In order to be not restricted to a linear dependence, $h_S$ is binned into 100 equidistantly spaced levels ranging from the global minimum to the global maximum surface elevation of the COSMO simulation domain. Each bin is assigned a binary field (one where $h_S$ falls into the respective bin, elsewhere zero). The linear model then provides a discrete but non-linear relationship in $h_S$. Similarly to $h_S$, the surface normal orientation is binned into

100 levels (10 levels each for $\phi_S$ and $\psi_S$). $\psi_S$ ranges from a value of zero to the domain maximum value. Cloud attenuation, considered in a linear dependency, is given by $\eta_{cl} = \exp(-\tau_{cl})$, with the cloud optical thickness $\tau_{cl} = k_L wp_L + k_I wp_I$. The absorption coefficients are assumed to be $k_L = 150\,\mathrm{m^2\,kg^{-1}}$ for liquid water and $k_I = 30\,\mathrm{m^2\,kg^{-1}}$ for cloud ice. $wp_L$ and $wp_I$ are the liquid and ice water paths of the entire atmospheric column, respectively. The linear model is set up by a matrix $\mathbf{M}$ with dimension sizes $n \times m$, with the number of rows $n$ equal to the COSMO domain size and the number of columns $m$ equal to the

number of independent variables. For the downscaling of $T_S$, the matrix is filled as follows: 4 columns contain the respective land cover fractions, 200 columns the binary fields associated with $h_S$ and ($\phi_S, \psi_S$), 2 columns the cloud attenuation and precipitation rate, and finally the last 9 columns the different powers of the horizontal coordinates in the bicubic interpolation function. The regression coefficients $a_S$ are obtained by minimizing

$$||\mathbf{M}a_S + fr^u T_S^u - T_S|| = !min. \tag{C1}$$





660 Here, $fr^u$ is the urban surface fraction, and $T_S^u$ is the associated temperature field. The downscaled field $T_S$ is then re-composed by multiplying $a_S$ with a new matrix $\tilde{\mathbf{M}}$, which is filled with respective data from the higher-resolved domain. The interpolated urban contribution $fr^u T_S^u$ is finally added to the result. The accuracy of the revised regression scheme is demonstrated in Fig. C1. Fig. C1a shows the temporal evolution of $T_S$ for the site DW located within the Dresden Basin, and the elevated site RW located on the northern rim of the basin (see Fig. 4 for a map overview). As can be seen, the approximated

665 $T_S$ series of the regression model (full lines) do not deviate much from the original COSMO $T_S$ series. In particular, it is shown that the temperature change with surface elevation is well represented. The deviations between the regression model and COSMO seem to be somewhat greater for the site RW, where the diurnal minimum $T_S$ is underestimated by $1 - 2\,\mathrm{K}$ with the regression model on days 3 and 4 of the simulation period. Fig. C1b shows a map of the temporal mean absolute error (MAE) for simulation period T2. As can be seen, MAE is typically around $0.5\,\mathrm{K}$. It is even lower for the urban sections within

670 the Dresden Basin, which is not surprising given that urban surfaces are excluded from the linear model. The temporal and domain average correlation coefficient of the regression model is $0.93$ for period T2.

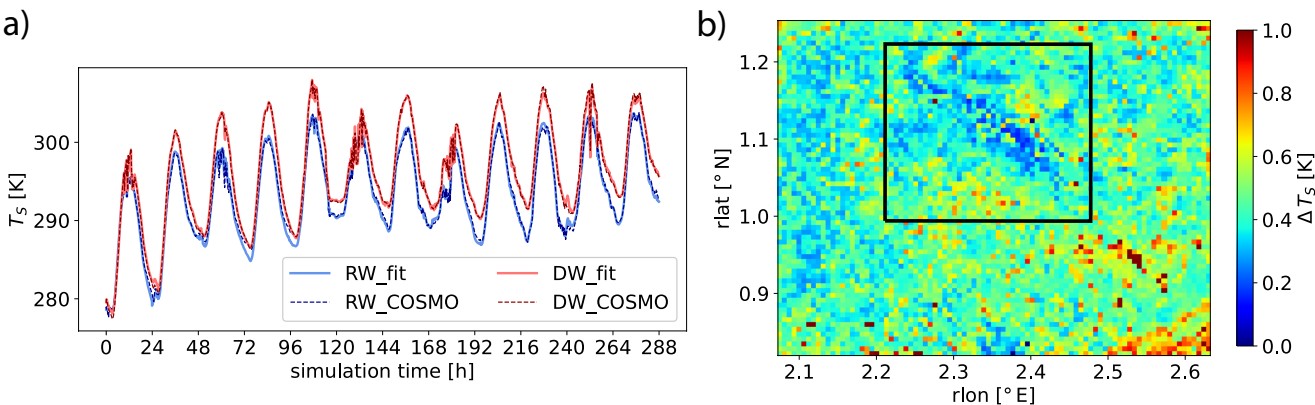

**Figure C1.** (a): Comparison of COSMO domain D3 surface temperature (dashed lines) with the values predicted by the multiple linear regression model (full lines) described in the main text of Section C1 during simulation period T2. The red lines refer to a location within the Dresden Basin (site DW) and the blue lines to a location on an adjacent hill (site RW). (b): Mean absolute error (MAE) of the regressed surface temperature over the domain D3 area during simulation period T2. The black box indicates the CAIRDIO domain D4.

### C2 Anelastic approximation, cloud microphysics, and radiative transfer

In our previous model version, the Boussinesq approximation for an incompressible fluid was used to consider buoyancy effects. The error introduced by the assumption of a constant-in-space air density restricted model applications to shallow

675 computation domains. This restriction is now alleviated with the use of the anelastic approximation, which uses a reference density vertical profile ($\rho_0 := \rho_0(z)$). In addition to $\rho_0$, the model requires air pressure $p$ for the optional computation of warm clouds and radiative transfer, which are not included in a previous model version and are further described below. In





contrast to $\rho_0$, $p$ may vary also horizontally. As with all other external meteorological fields, $\rho_0$ and $p$ are derived from the mesoscale fields of the driving COSMO simulation. In the previous Section C1, a multiple linear regression model to spatially
downscale, amongst other surface variables, the surface pressure field $p_S$ is presented. Using reconstructed $p_S$ together with the spatially (trilinear) interpolated virtual temperature field $T_v$ of COSMO, the downscaled fields $p$ and $\rho_0$ are obtained by vertical integration of the hydrostatic balance, starting from the surface $h_S$:

$$p(z) = p_S - \int_{h_S}^{h_S+z} g\rho(z)\mathrm{d}z, \tag{C2}$$

$$\rho(z) = \frac{p(z)}{R_d T_v}, \tag{C3}$$

where $g = 9.81\,\mathrm{m\,s^{-1}}$ is the gravity acceleration and $R_d = 287\,\mathrm{J\,kg^{-1}K^{-1}}$ the ideal gas constant of dry air. Note that Eq. C2 is applied to each vertical column of the CAIRDIO grid separately, so $\rho$ still requires horizontal averaging along planes of $z' = z + h_S = \mathrm{const.}$ to obtain $\rho_0$.

For the simulation of low-altitude clouds and fog formation at temperatures above the freezing point of water, a single-moment bulk warm-cloud microphysics scheme was implemented. This scheme uses only cloud liquid water and rain water as cloud and
precipitation constituents. Relevant microphysical processes include condensation/evaporation of cloud water, auto-conversion and collection of cloud water to form rain water, and evaporation and sedimentation of rain water. The bulk parameterizations treat these processes in terms of the mass-mixing ratios of water vapor $q_v$, cloud water $q_c$, and rain water $q_r$. The parameterizations are basically adopted from Klemp and Wilhelmson (1978). The cloud microphysics scheme is applied at each model time step after the integration of all other prognostic tendencies (advection, diffusion, ecc.) of the meteorological variables.
The first process to be treated in the microphysics scheme is rain formation:

$$\partial_t q_r^{form} = \max[k_1(q_c - q_c^t)\,0] + k_2 q_c q_r^{0.875}, \tag{C4}$$

with $k_1 = 10^{-3}$, $k_2 = 2.2$, and the threshold cloud water mixing ratio for auto conversion $q_c^t = 5 \times 10^{-4}$.

Thereafter, in general, $q_v$ and $q_c$ can be out of equilibrium, i.e. $q_v$ may either exceed saturation with respect to liquid water $q_v^s$, or clouds may be present ($q_c > 0$) in sub-saturated conditions $q_v < q_v^s$. The condensation/evaporation rate to restore ther-
modynamic equilibrium is computed by a saturation adjustment technique, which numerically solves the following equation for $\delta T$ using Halley's $3^{\mathrm{rd}}$ order method:

$$L\delta T = q_v - \frac{p_v^s[T + \delta T]}{p - p_v^s[T + \delta T]}\frac{R_d}{R_w}. \tag{C5}$$

Here, $L = 2.257 \times 10^6\,\mathrm{J\,kg^{-1}}$ is the heat of evaporation, $T = \Theta(p/p_0)^\kappa$ the prognostic model temperature using interpolated pressure $p$, $p_0 = 1.013e5\,\mathrm{Pa}$, $\kappa = 2/7$, and $R_w = 462\,\mathrm{J\,kg^{-1}\,K^{-1}}$ the gas constant of water vapor. The saturation vapor
pressure $p_v^s$ is computed according to an updated formula of Arden Buck:





$$p_v^s[T] = 611.22 \exp\left[\left(18.678\,\mathrm{K} - \frac{T - 273.15\,\mathrm{K}}{234.5\,\mathrm{K}}\right)\left(\frac{T - 273.15\,\mathrm{K})}{T - 16,01\,\mathrm{K}}\right)\right] \tag{C6}$$

The condensation/evaporation rate then follows by

$$\partial_t q_c^{cond} = \frac{\max\left(\delta T L, -q_c\right)}{\delta t}. \tag{C7}$$

The last microphysical process considered is rain evaporation, whose parameterization is also taken from Klemp and Wilhelmson (1978):

$$\partial_t q_r^{evap} = \frac{1}{\rho_0} \frac{(q_v/q_v^s - 1)[1.6 + 124.9\,(\rho_0 q_r)^{0.2046}](\rho_0 q_r)^{0.525}}{5.4 \times 10^5 + 2.55 \times 10^6 (p q_v^s)^{-1}}, \tag{C8}$$

where $\rho_0$ and $p$ are provided in units of $\mathrm{g\,cm^{-3}}$ and $\mathrm{hPa}$, respectively. Note that since Eq. C8 is applied after the saturation adjustment it can be only effective in sub-saturated conditions. The sedimentation of rain droplets is not directly considered in the microphysics scheme but is accounted for by an additional vertical advection step following the 3-D standard advection routine for $q_r$. As the terminal fall velocity, $v_r$ can significantly exceed the average wind speed, this advection step is split into a variable number of sub-steps in order to not impair model stability or integration efficiency.

The realistic simulation of low-altitude stratiform clouds requires the consideration of radiative processes, in particular the enhanced radiative cooling of the cloud tops. For this purpose, an external library of the Rapid Radiative Transfer Model (RRTM) (Mlawer et al., 1997) is used. RRTM is based on the two-stream approximation and computes heating rates for long- and short-wave interactions within each layer of an atmospheric column. Aside from the vertical profiles of air pressure and temperature, RRTM requires profiles of the atmospheric gaseous composition. The contributions of the greenhouse gases methane, nitrous oxide, ozone, and a multitude of hydrofluorocarbon species are considered by a spatially constant effective carbon dioxide molar mixing ration, which is assumed to be $500\,\mathrm{ppmv}$ and thus higher than the current average value of about $410\,\mathrm{ppmv}$. As non-gaseous constituents, RRTM considers aerosols and clouds. Clouds are represented by the mixing ratios of liquid water and cloud ice. The radiative-transfer equations are solved for the entire vertical extent of the troposphere and lower stratosphere, but of the required input data for clouds, only $q_c$ is provided with high spatial resolution within the vertical height range of the CAIRDIO computation domain. Therefore, ice clouds and other liquid-water clouds above the domain top are interpolated from the hosting COSMO simulation with lower spatial and temporal accuracy. Similarly, profiles of pressure, specific humidity, and temperature are filled-in with COSMO data above the CAIRDIO domain top.

*Author contributions.* Michael Weger contributed in study conceptualization, execution and evaluation of model runs, and paper writing. Bernd Heinold supervised the study, provided technical support, and contributed in paper proof reading.





*Competing interests.* All authors declare that they have no competing interests.

*Acknowledgements.* Recent German-wide emission data were provided by the German Environment Agency (Umweltbundesamt, UBA).
A special thanks goes to Uwe Wolf from the Saxon State Office for Environment, Agriculture and Geology (Sächsisches Landesamt für Umwelt, Landwirtschaft und Geologie, LfULG) for providing line emissions for road transport in Dresden. Measurements of BC and other meteorological variables from the consulted air-monitoring sites were provided by the LfULG. Building geometries and orography (DGM1) are available from the State Enterprise for Geographic Information and Surveying Saxony (GeoSN). We thank the DWD for good cooperation and support.



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
