# Peer review of "Air pollution trapping in the Dresden Basin from gray-zone scale urban modeling."

_Atmospheric Chemistry and Physics, 2023_

## Author Comment (AC1)

**Response to comments of Referee 1**

Specific comments:

1)      Title

In the title the emphasis is given both to natural and urban topography. The effect of urban topography is taken into account by the microscale model, but it is not discussed in the paper. For this reason, the Authors should either improve the discussion of the effects of urban topography on air pollutant dispersion or remove urban topography from the title.

It is true that urban effects are not really discussed in our manuscript as they were already extensively discussed in our previous simulation study. Therefore, we decided to change the title as suggested by another Referee. The new title is: "Air pollution trapping in the Dresden Basin from urban gray-zone scale modeling."

2)      Paper organization

I found that the Section on principal-component analysis is not well connected with the rest of the paper. In particular, it is not clear what is the added value of this additional analysis in characterizing the effect of topography on pollutant dispersion in the Dresden basin. Therefore, the motivation and the added value of this additional analysis should be better explained.

We agree that this Section was not well connected to the rest of the paper. First of all, we removed the more technical part of the principal-component wind field analysis, as it is already covered in the work by Ludwig et al. (2004). As a result, the motivation of the principal component analysis is now better explained at the beginning of Section 3.4. We also revised other parts of this Section. For example, we focus the discussion more on the BC age concentration patterns instead of the mass concentration patterns, because we think that differences in the transport can be better seen in the age concentration.

Minor and technical remarks

Line 60 and line 66: use lowercase after colons.

Corrected.

Line 81: "mean age of stratospheric age": repetition.

Corrected.

Line 109: "for a MORE realistic simulation…"

Changed accordingly.

Equation 1: c is not introduced.

The definition of c is now included.

Equation 3: $c_a$ is not introduced.

We replaced "α" with $c_a$ for the age concentration in the whole manuscript.

Line 247: delete the full stop after "planes".

Deleted.

Caption Figure 4: change "Section A" with "Appendix A".

Changed accordingly.

Title Section 3.1.2: change "March" with "May".

Corrected.

Some Section titles are repeated: "Age-concentration modeling – proof of concept", "Principal-component analysis".

Repeated titles are removed.

Line 466: "with generally lower wind speeds than in cases…" probably something is missing in this sentence.

This half sentence was removed.

Line 498: change "vise versa" with "vice versa".

Corrected.

Line 536: I would change "reflects the real distribution of air pollutant sources" with something similar to "reflects the presence of accumulation or recirculation areas".

We replaced it with "The age concentration can therefore be considered a more appropriate metric that reflects the accumulations from a realistic distribution of air pollutant sources."

Figure 5: it would be interesting to have also a map of the total emissions, and not only of the emissions within the first vertical layer.

All the emissions from the elevated layers are now shown in a second plot. By the way, we decided to move this Figure to an extra supplemental file (together with some other figures), because it is not really discussed in the manuscript.

Figure7 and 8: in my opinion for the interpretation of the results it would be clearer to plot the time series of wind speed and direction and not of the two horizontal wind components.

We agree therein and changed the Figures accordingly.

---

## Author Comment (AC2)

**Response to comments of Referee 2**

**General comments:**

It is a little bit bothering me when you use the term "age" in the manuscript. In my understanding, aerosol's degree of aging should be referred to not only the lifetime in the atmosphere but also how many reactions they involved (both physical and chemical). Or do you simply refer to how many days after aerosol is generated? Could you make it clear in the manuscript?

We agree that there has been some ambiguity regarding the definition of "age" in the introduction of the manuscript. In our case, with "age" we refer to the atmospheric residence time without regard to any chemical or physical aerosol processes. This was now clarified in the introduction (line 69). We decided to keep the term "age" in the manuscript as it is shorter and catchier than "atmospheric residence time". We can further refer to the work of Deleersnijder and Campin (2001), who also used the term "age" to describe particle residence times.

For equation 2, I do not understand why you remove the source and sink for aging. The aging process of BC in the atmosphere is very complicated (see "Bounding the role of black carbon in the climate system: A scientific assessment"). I suggest adding some discussion about the atmospheric aging processing of BC and why you decide not to include that.

In Section 2.1.1 we included a brief discussion of what simplifications are implicitly assumed with regard to the source and sink terms in equation 2. It is true that we neglected additional source terms that would arise if physicochemical aging processes (e.g., hydrophilic coating) and therefore multiple BC sub-species were considered. Including this additional complexity would, however, be beyond the scope of our paper, which considers BC as a suitable tracer to investigate local air pollutant transport.

This manuscript is too long. Since it is an extension of a previously published work, maybe you can remove common parts and refer to that paper.

We shortened the technical description of the simulation setups in Sections 2.2.1 and 2.2.2 as indeed much of this content is already included in our previous paper. We kept only the parts that are more specific to the present case study.  Furthermore, we removed the detailed description of the EOF

methodology in Section 3.4 as it does not fit well into this chapter. Lastly, we decided to create a supplementary pdf file for figures that are not central to the written paper content.

For your equations, could you define all parameters clearly and put their unit? Maybe you can make a table.

We added a Table at the end of the manuscript with all relevant mathematical symbols. Moreover, we checked the entire manuscript to ensure that variables are defined everywhere consistently. We also changed some definitions, e.g., the age concentration is now defined by c_a and not alpha.

When you discuss other models for estimating aerosol aging, I suggest adding a discussion of the FLEXPART and CO tracer method since a recently published paper ("Particle phase-state variability in the North Atlantic free troposphere during summertime is determined by atmospheric transport patterns and sources") used this method to estimate the aging of the aerosol air mass.

We added the FLEXPART part model in our introduction where we introduced the Lagrangian particle dispersion models.

**Specific comments:**

I suggest improving the title by mentioning this is a model study and the model name since the current title sounds like a field measurements

The title was changed to "Air pollution trapping in the Dresden Basin from gray-zone scale urban modeling." This title reflects that it is a modeling study and also mentions the addressed spatial scale.

L144-150, "The first two … of their age." You can either mark the terms in the equation or show the terms in the manuscript. It is not very clear to me when I read it.

We added some text labels to the individual terms in the equations for more clarity.

L272-273, "For the air-chemistry … 14km resolution." Could you explain what you mean by air chemistry? Do you mean chemical reactions in the gas phase

or aerosol phase or both (heterogeneous reactions)? Moreover, what reactions you included (e.g., ozonation, aqueous phase reaction, photolysis, NOx reaction, etc.,).

We decided to shorten this part of the manuscript and refer to our previous manuscripts where the mesoscale setup and used air-chemistry mechanism are explained in more detail. Anyways, it was a bit misleading to write air chemistry, because BC is assumed to be chemically inert in our model.

L275, "Domains ... respectively." How do you choose the location and size of D1 to D3? Moreover, I think you should not use the resolution since that means how many ticks per length. Maybe you should use distance.

We replaced "horizontal resolution" with "horizontal grid spacing" as this is exactly what is meant by it. Domain sizes are chosen rather arbitrarily but it is ensured that the relaxation fetch between the inner and outer domain boundaries at each nesting step is sufficient for a relaxation of the lateral boundary conditions. The locations are chosen such that the target area is located close to the domain centers.

Figure 3. Where is D4? It is not clear to me. Also, I think you should label D2. It is not clear to me which box is D1 since D1 is labeled between two boxes.

Figure 3 was improved and split into two plots in order to better show domains D2, D3, and D4. We also used different colors for the boxes such that the relationship between the boxes and labels should be clear now.

Figure 6. I suggest using a color different from the color bar for the mark of Dresden. Moreover, do you have any spots that have MSL pressure below 996 hPa? If not, you can reduce the range of the color bar to cover the min and max of pressure.

We changed the colormap of the figure, adjusted the color of the marks, and restricted the color scale to 996 hPa.

Figure 7. It is not clear to me how you calculate the b and c color bars and why b and c have different Z ranges.

Maybe it was confusing that we placed the color bar labels of (b) and (c) not directly to the left of the color bar but on top of the plot. This was changed accordingly, such that it is clear that (b) shows $u_{||}$ and (c) shows the vertical gradient in virtual potential temperature. We also explained more clearly how $u_{||}$ is defined. We also equalized the z ranges of plots (b) and (c) for better comparison.

Figure 8. I suggest adding a caption even if it will be the same as Figure 7's.

We also included a more detailed caption in Figure 8.

L380-381, "3.3 ... concept." Please check all your section numbers.

The double section was removed.

Figure 12. What is the color bar?

"The color bar indicates the horizontal wind speed of the reconstructed wind field patterns in (c-f)." was added to the caption of Figure 12.

---

## Author Comment (AC3)

**Response to comments of Referee 3**

Specific comments:

Introduction: it is unclear as to why the Dresden Basin is an important region to study for the Special Issue. There are also no discussions on the health impacts of air pollution in this region. Observations from either satellite or ground-based measurements should be addressed to support the significance of air pollution in this region.

We primarily decided to study the Dresden Basin with our model, because not much is known about air pollution in this rather small urban basin. The Dresden Basin can be considered to be too small to resolve the meteorological aspects relevant to air pollution dispersion with a mesoscale model. On the other hand, it is still too extensive to feasibly cover it with classic urban microscale simulations. Therefore, it is an interesting target for our microscale/urban gray zone model.  From theoretical considerations, it can be expected that under stable weather conditions, air quality can substantially degrade within this basin, because it is mostly urban, rather small, and also well-secluded as the downstream exit is very narrow. Another important factor is that the Dresden Basin is connected to the much larger and often heavily polluted Most Basin located up the valley in the Czech Republic. To better support the significance of air pollution in this region, we further include Sentinel 5P satellite NO2 observations in the supplemental pdf file of the manuscript and refer to it in the introduction.

Figure 3. The map does not indicate the absolute coordinates of the study region, so readers who are unfamiliar with the region would not be able to efficiently locate the exact region. I suggest that the authors either add a separate map to show the geographic location of Dresden or modify the coordinates of the current map to absolute coordinates. Also, "hmsl" needs to be defined.

We revised Figure 3 to include domain D4 in the nesting chain. Furthermore, the absolute geographical coordinates are now shown instead of the coordinates of the rotated-pole lat/lon grid (on which the mesoscale simulations were actually performed). We replaced "hmsl" with "$z_{asl}$", which means height above sea level (also defined now in the caption of Figure 3.)

P33 Line 724: It is unclear how aerosols are represented in the RRTM since they will affect the radiative fluxes. It would be helpful to readers if the authors describe the aerosol optical properties if the air pollution in the Dresden Basin in terms of single scattering albedo, asymmetry parameter and Ångström exponent.

This was actually a mistake in our description. RRTM can consider aerosol and clouds, but we considered only cloud radiative effects in our simulations. Considering that cloud radiative effects from fog /low-altitude clouds are of orders of magnitude more important than boundary-layer aerosol effects, we think this is a valid first-order approximation. However, clearly, air pollution within the boundary layer has a significant radiative effect too. But in order to consider it realistically, other aerosol types besides BC (e.g., sulfate) have to be included in the simulation too, which would require air chemistry in our microscale simulations.  This is, however, not yet included in our actual model version.

P2 Line 37: Remove "In fact"

Removed.

Figure 2: "in experiment two" should be a new sentence

Changed accordingly.

Sections 3.3 and 3.5 do not have contents. Please fix the section numbers of the whole manuscript.

Double sections are removed.